# New insight into the molecular mechanism of colour differentiation among floral segments in orchids

Bai-Jun Li [1], Bao-Qiang Zheng[1], Jie-Yu Wang[2], Wen-Chieh Tsai[3,4], Hsiang-Chia Lu[5], Long-Hai Zou [1,6], Xiao Wan[1,7], Di-Yang Zhang[5], Hong-Juan Qiao[1], Zhong-Jian Liu [5,8✉] & Yan Wang [1✉]

An unbalanced pigment distribution among the sepal and petal segments results in various colour patterns of orchid flowers. Here, we explored this type of mechanism of colour pattern formation in flowers of the *Cattleya* hybrid 'KOVA'. Our study showed that pigment accumulation displayed obvious spatiotemporal specificity in the flowers and was likely regulated by three R2R3-MYB transcription factors. Before flowering, *RcPAP1* was specifically expressed in the epichile to activate the anthocyanin biosynthesis pathway, which caused substantial cyanin accumulation and resulted in a purple-red colour. After flowering, the expression of *RcPAP2* resulted in a low level of cyanin accumulation in the perianths and a pale pink colour, whereas *RcPCP1* was expressed only in the hypochile, where it promoted α-carotene and lutein accumulation and resulted in a yellow colour. Additionally, we propose that the spatiotemporal expression of different combinations of *AP3-* and *AGL6-*like genes might participate in KOVA flower colour pattern formation.

[1] State Key Laboratory of Tree Genetics and Breeding; Research Institute of Forestry, Chinese Academy of Forestry, 100091 Beijing, China. [2] Key Laboratory of Plant Resources Conservation and Sustainable Utilization, South China Botanical Garden, Chinese Academy of Sciences, 510650 Guangzhou, China. [3] Institute of Tropical Plant Sciences and Microbiology, National Cheng Kung University, 701 Tainan City, Taiwan. [4] Orchid Research and Development Center, National Cheng Kung University, 701 Tainan City, Taiwan. [5] Key Laboratory of National Forestry and Grassland Administration for Orchid Conservation and Utilization at College of Landscape Architecture, Fujian Agriculture and Forestry University, 350002 Fuzhou, China. [6] State Key Laboratory of Subtropical Silviculture, Zhejiang A & F University, 311300 Lin'an, China. [7] Research & Development Center of Flower, Zhejiang Academy of Agricultural Sciences, 311202 Hangzhou, China. [8] College of Forestry and Landscape Architecture, South China Agricultural University, 510642 Guangzhou, China. ✉email: zjliu@fafu.edu.cn; wangyan@caf.ac.cn

Flower colour patterns are generally shaped by colour differentiation in the petal segments and can help plants attract pollinators[1,2], in addition to being an important ornamental trait[3]. Therefore, the exploration of the mechanism of colour differentiation is valuable for understanding plant evolution and breeding novel ornamental lines. For most species, the sepals are just uniformly green and do not contribute to interesting colour patterns. Orchids are exceptional in that the sepals and the lip are usually as colourful as the petals, which results in an unbalanced pigment distribution among different segments of the perianths and lip to show various flower colour patterns. Although numerous studies have reported the mechanisms of pigment biosynthesis in flowers[4–7], the formation of colour patterns by different types of pigments among the sepal and petal segments has yet to be systematically described. *Rhyncholaeliocattleya* Beauty Girl 'KOVA' (KOVA) is a well-known ornamental orchid cultivar; its flowers are relatively large and exhibit pale pink perianths (sepal/petal), a purple-red epichile and a yellow hypochile (Fig. 1a). The red and yellow colours are determined by anthocyanins and carotenoids, respectively (Fig. 1b). These characteristics make KOVA a suitable target for addressing the interesting and important question of the mechanism of colour pattern formation.

Anthocyanins and carotenoids are widely distributed pigments in flowers. The anthocyanin biosynthesis pathway (ABP) is highly conserved and has been well studied in plants[6]. As crucial ABP regulators, the R2R3-MYB transcription factors can regulate the expression of structural genes to influence flower colour formation[8–13]. In Orchidaceae, some R2R3-MYBs have been verified to activate ABP structural genes to promote anthocyanin accumulation in *Phalaenopsis*[5], *Oncidium*[14] and *Dendrobium*[15] flowers. Carotenoids are another important type of pigment that contributes to colouration ranging from yellow to red in plant tissues[16,17]. However, the R2R3-MYBs involved in the regulation of the floral carotenoid biosynthesis pathway (CBP) are not well known. Recently, *RCP1* (*R2R3-MYB*) was suggested to activate CBP structural genes and increase carotenoid accumulation in *Mimulus lewisii* flowers[18].

The shapes and colour patterns of flowers are used as visual standards of flower organ identity. According to the ABCDE model, different combinations of MADS-box genes determine flower organ identity[19]. B-class genes, including *APETALA3* (*AP3*) and *PISTILLATA* (*PI*), and E-class *SEPALLATA* (*SEP*)-like genes are considered to participate in petal formation[20–22]. Many studies have shown that the overexpression or silencing of *B-/E*-class MADS-box genes can change the shape of petals[20–23]. Notably, as the transcription levels of these MADS-box genes change, petal colours are also altered[23–30]. Previous studies have indicated that combinatorial protein interaction networks among the members of the *B*-class, *E*-class and *AGL6* clades during orchid floral development might be responsible for the evolutionary novelties of orchid flowers[24,31–33]. Recently, the P code was proposed, indicating that OAP3-1 and OAGL6-1 are required for perianth formation, whereas OAP3-2 and OAGL6-2 are necessary for lip

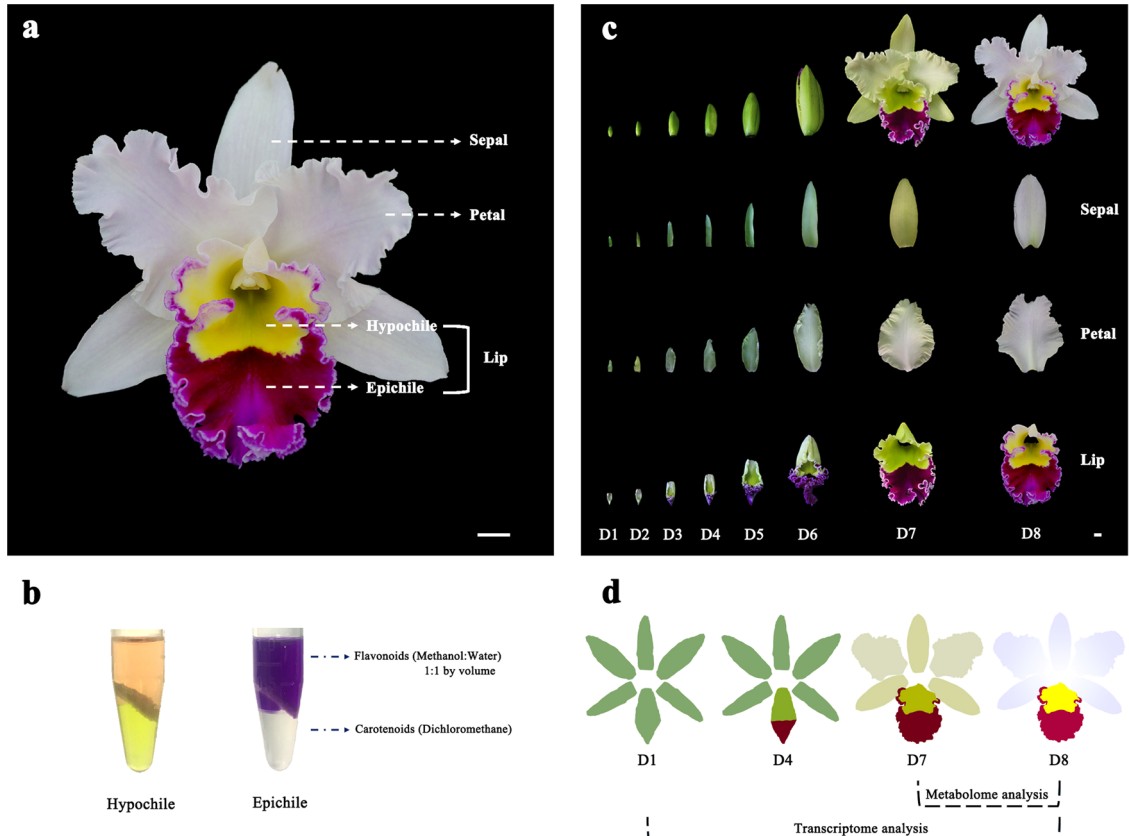

**Fig. 1 Characteristics of *Rhyncholaeliocattleya* Beauty Girl 'KOVA' (KOVA) flowers. a** KOVA flower with pale pink perianths, a purple-red epichile, and a yellow hypochile in the full-bloom stage. **b** The epichile is pigmented by anthocyanins, and carotenoids are the main contributor to the yellow colour in the hypochile. **c** Morphological anatomy of KOVA flowers during floral development. Development stage 1 (D1, bub length < 2 cm), D2 (bub length 2–3 cm), D3 (bub length 3–4 cm), D4 (bub length 4–5 cm), D5 (bub length 5–6 cm), D6 (bub length > 6 cm), D7 (2 days after flowering), and D8 (ten days after flowering). **d** Sampling strategy: sepals, petals, epichiles, and hypochiles were collected at D1, D4, D7 and D8, respectively, for transcriptome sequencing and metabolome identification. Bar = 1 cm.

development in orchids[23]. However, little is known about the relationship between MADS-box genes and flower colour pattern formation in orchids and other plants.

In this study, based on the identification of large-scale metabolomes, *RNA-Seq* sequencing, and *Iso-Seq* sequencing, we revealed the precise spatiotemporal characteristics of colour pattern formation during KOVA floral development. Then, we identified two R2R3-MYBs that might participate in the regulation of the ABP and one that might be involved in regulating the CBP in flowers. In addition, we propose that *AP3-* and *AGL6-*like genes might participate in colour differentiation in the perianth and lip segments based on the analysis of transcriptome data and a peloric mutant of KOVA.

## Results

**Morphological anatomy of KOVA flowers during development**. The analysis of morphological anatomy showed that no red or yellow colouration appeared in the perianths and lip before floral development stage 2 (D2; Fig. 1c). From D2 to before D7, the colour of the perianths and hypochile changed from green to yellow green, and gradually increasing purple-red colouration appeared in the epichile. After flowering (i.e., from D7 to D8), the colour of the perianths changed from yellow green to white with pale pink; the hypochile turned yellow; and the epichile remained purple-red, but the degree of this coloration from D7 to D8 was lighter than that in previous stages. Based on the analysis of the morphological anatomy, we collected sepals, petals, epichiles, and hypochiles at D1, D4, D7 and D8, respectively, for transcriptome sequencing and metabolome identification (Fig. 1d).

**Identification and quantification of pigment components**. Liquid chromatography-electrospray ionisation-tandem mass spectrometry (LC-ESI-MS/MS) analysis identified eleven carotenoids and was used to quantify their levels in flowers at D7 and D8 (Figs. 1d and 2a). The total carotenoid content of the hypochile was higher than that of other flower segments at both stages (Fig. 2a). At D8, α-carotene and lutein were the main carotenoids in the hypochile, accounting for ~31 and 45% of the total carotenoid content, respectively. Moreover, cyanidin and the four forms of cyanin were detected in all flower segments (Fig. 2a). The total anthocyanin content of the epichile was highest at D7 and D8. Among the cyanins, the amount of cyanidin O-acetylhexoside was highest in all flower segments. Additionally, the total chlorophyll content of all segments at D7 was obviously higher than that at D8, and the total chlorophyll content in the perianths was higher than that in the epichile and hypochile at D7 (Fig. 2a). The total carotenoid, anthocyanin, and chlorophyll contents decreased in all flower segments from D7 to D8. These results suggested that cyanins produced via the ABP and lutein and α-carotene synthesised via the CBP contributed to the coloration of KOVA flowers ranging from purple-red to yellow, while chlorophyll provided green colouration.

**Expression profile of structural genes in pigment biosynthesis pathways**. Through PacBio *Iso-Seq* sequencing, 17.61 gigabytes (GB) of raw data were produced from two libraries (0–4 and 4–10 kb libraries), and 67,113 unique full-length transcripts (isoforms) were obtained through pipeline analysis. Moreover, 1.894 billion pairs of clean reads were produced from 48 libraries using *RNA-Seq* to calculate the expression level (fragments per kilobase per million, FPKM) of each isoform, with a mapping ratio of 77.04–85.99%. Additionally, the unmapped reads were *de novo* assembled, and 77,281 unigenes were produced.

The isoforms/unigenes involved in the ABP and CBP were screened from the transcriptome data-set, and heatmap analysis was used to illustrate the expression profiles of ABP and CBP structural genes in flower segments during floral development based on their FPKM values. The results showed that the expression of all ABP structural genes was barely detectable at D1, which is consistent with the absence of red colour in KOVA flowers (Fig. 2b). Thereafter, the expression of the structural genes involved in the ABP was strongly activated only in the epichile at D4, at which time the epichile displayed an obvious purple-red colour. Notably, all components of the ABP were expressed in the perianths and epichile at D7, although the expression level of ABP structural genes in those at D7 was lower than that in the epichile at D4. At the final stage, the expression of ABP structural genes was decreased in all segments, and the perianths and epichile displayed pale pink and lighter purple-red colouration, respectively (Fig. 1a). The dynamic expression of ABP structural genes in different segments during floral development is consistent with the morphological anatomy of the purple-red colouration that appears in the epichile before flowering and the pale pink colour that appears in the perianths after flowering (Fig. 1c). Furthermore, the ABP was strongly activated in the epichile and weakly activated in the perianths, which supported the metabolomic results showing that higher levels of cyanins accumulated in the epichile than in the perianths (Fig. 2a).

On the other hand, the expression of all CBP structural genes was downregulated in all segments at D1, except for the *lycopene ε-cyclase* (LCYE) and *lycopene β-cyclase* (LCYB; Fig. 2c). At D4, in addition to the *LCYE*s and *LCYB*s, *β-carotene hydroxylase* 1 (*BCH1*) was activated only in the hypochile, while *BCH2*s were expressed in all flower segments. At D7, the hypochile was yellower than the other segments, and the expression of the *phytoene synthase* (PSY) was obviously upregulated only in the hypochile. However, the upstream *phytoene desaturase* (PDS), *carotenoid isomerase* (CRTISO), *BCH2* and *P450-type β-ring hydroxylase* (CYP97A3) were obviously upregulated in all segments, and the *BCH1*s were upregulated in the hypochile. Additionally, the expression of the *LCYE*s was downregulated in all segments, and the expression level of the *LCYB*s decreased only in the hypochile. In the final stage, only the hypochile displayed a yellow colour; all of the upstream CBP genes from the *PSY*s to the *CRTISO*s were upregulated; and the *LCYB*s and *LCYE*s were downregulated in all segments. Notably, the *BCH1*s remained highly expressed in the hypochile, whereas *BCH2*s and *CYP97A3*s were downregulated in all segments (Fig. 2c). Most CBP structural genes, especially the *PSY*s and *BCH1*s, remained upregulated only in the hypochile after flowering, which supported the results of the morphological anatomical and metabolomic analyses indicating that the carotenoid content was highest in the hypochile among the flower segments, resulting in a yellow colour (Figs. 1c and 2a).

Moreover, two types of *carotenoid cleavage dioxygenase* (CCD), the *CCD1* and *CCD7*, were identified from the transcriptome data. These genes began to show upregulation in D7, and their expression reached its highest level at D8 (Fig. 2c), whereas the carotenoid contents of the segments decreased from D7 to D8 (Fig. 2a). However, the *CCD1*s were expressed in all segments at D8, whereas the expression levels of the *CCD7*s remained higher only in the perianths and epichile.

**Identification of *R2R3-MYBs* involved in the regulation of pigment biosynthesis**. The maximum likelihood (ML) phylogenetic tree suggested that eleven R2R3-MYBs from KOVA belonged to subgroup 21 of the R2R3-MYBs, which is related to RCP1, a positive regulator of the CBP (Fig. 3a; the blue circle in

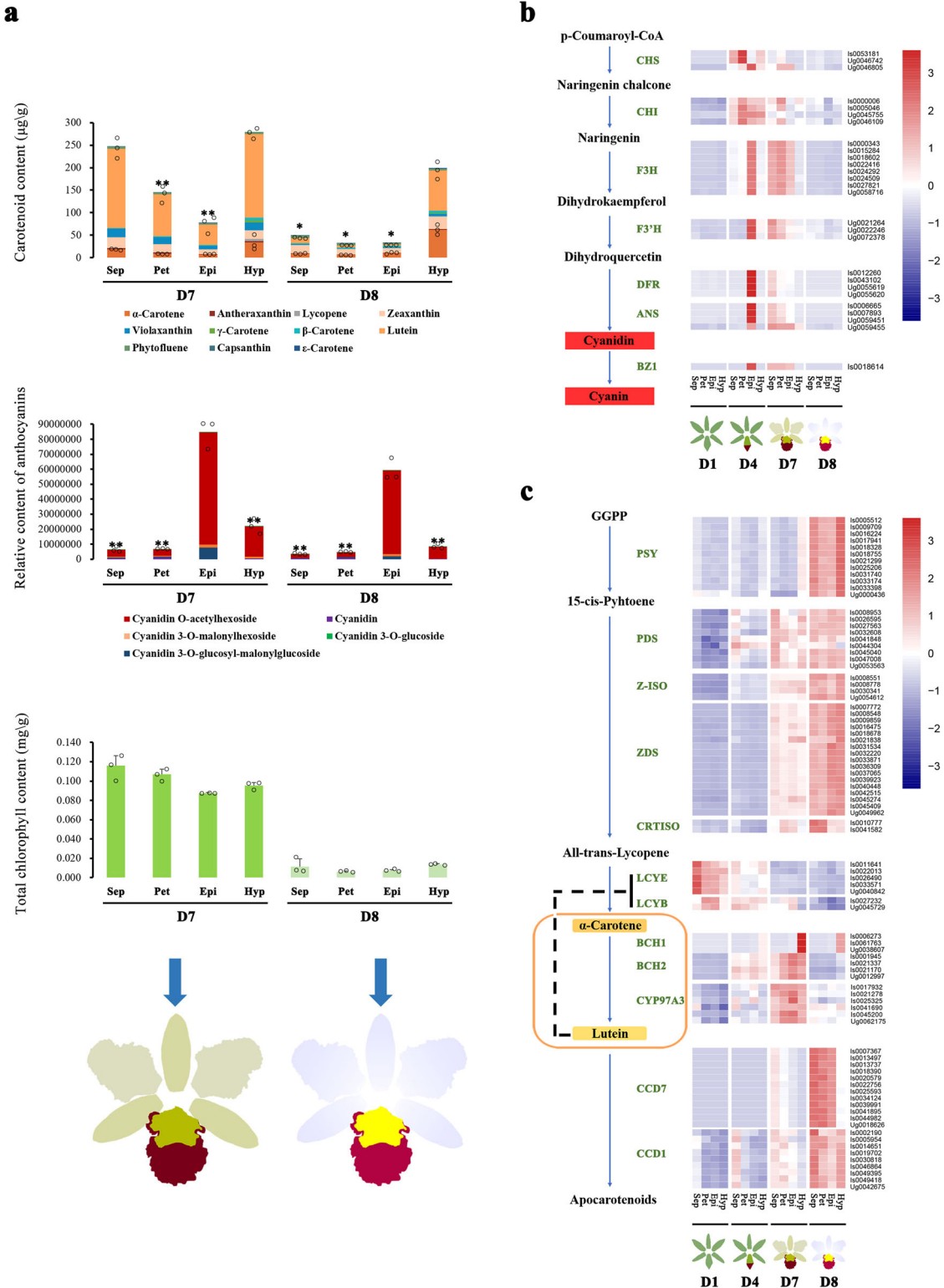

the figure). Additionally, two R2R3-MYBs belonged to subgroup 6, which is closely related to PeMYB2, a positive regulator of the ABP (Fig. 3a; the red circle in the figure).

Based on the FPKM values of the R2R3-MYBs of KOVA that belonged to subgroup 6 and subgroup 21, the heatmap analysis showed that the R2R3-MYBs of KOVA that clustered in subgroup 21 could be characterised as four types: type I R2R3-MYBs were upregulated in the hypochile at D7 and in all segments at D8; type

II R2R3-MYBs were upregulated only in the hypochile, and their expression level was highest at D7; type III R2R3-MYBs were upregulated only in the hypochile at D1; and type IV R2R3-MYBs were upregulated in all segments at D8 (Fig. 3b). Additionally, in subgroup 6, two different expression profiles of the R2R3-MYBs of KOVA were observed: type I R2R3-MYBs were upregulated only in the epichile at D4, and type II R2R3-MYBs were upregulated in the perianths and epichile at D7 (Fig. 3b). The

**Fig. 2 Identification and quantification of the pigments in KOVA flowers at D7 and D8. a** Eleven carotenoids and four cyanins were identified and quantified from the sepals, petals, and lip segments at D7 and D8 by using an LC-ESI-MS/MS system. The highest carotenoid and cyanin contents were detected in the hypochile and epichile, respectively. From D7 to D8, the total chlorophyll content was reduced in all segments. Sepal, petal, epichile, and hypochile are abbreviated as Sep, Pet, Epi, and Hyp, respectively. The data are the mean ± SD from three biological replicates. Asterisk and double asterisk indicate values that differ significantly from those in the segment with the highest pigment content in every stage at $P < 0.05$ and $P < 0.01$, respectively, according to Student's paired $t$-test. **b** The expression profile of ABP structural genes in the sepals, petals, and lip segments from D1 to D8. **c** The expression profile of CBP structural genes in the sepals, petals, and lip segments from D1 to D8. The data are the mean of three biological replicates. Isoform and unigene are abbreviated as Is and Ug, respectively. CHS Chalcone synthase; CHI chalcone isomerase; F3H flavanone 3-hydroxylase; F3′H flavonoid 3′-hydroxylase; DFR dihydroflavonol 4-reductase; ANS anthocyanidin synthase; BZ1 flavonol glucosyltransferase; PSY phytoene synthase; PDS phytoene desaturase; Z-ISO ζ-carotene isomerase, ZDS ζ-carotene desaturase; CRTISO carotenoid isomerase; LCYE lycopene ε-cyclase; LCYB lycopene β-cyclase; BCH β-carotene hydroxylase; CYP97A3 P450-type β-ring hydroxylase; CCD carotenoid cleavage dioxygenase.

expression profiles of the type I and II $R2R3$-$MYB$s were consistent with the expression profiles of ABP structural genes in the flower segments at D4 and D7, respectively (Fig. 2b).

To further confirm the potential functions of these $R2R3$-$MYB$s during flower development, we identified seventeen coexpressed gene modules and determined which $R2R3$-$MYB$s were coexpressed with structural genes involved in the ABP and CBP using weighted gene co-expression network analysis (WGCNA; Supplementary Fig. 1). In the midnight-blue module, the type II R2R3-MYB belonging to subgroup 21 (Unigene0031636) was coexpressed with the $BCH1$s (Fig. 3c). Additionally, we found that the type I R2R3-MYB belonging to subgroup 6 (Unigene0056193) presented a co-expression relationship with ABP structural genes in the light-yellow module (Fig. 3d).

The results of the phylogenetic tree analysis, heatmap analysis and WGCNA suggested that the expression pattern of Unigene0031636 is positively related to the expression of CBP structural genes during flower development and that this unigene is a candidate for the activation of the expression of CBP structural genes; it was designated *Rhyncholaeliocattleya* promoted carotenoid pigmentation 1 (*RcPCP1*). In contrast, Unigene0056193 is a candidate for promoting the expression of ABP structural genes and was designated *Rhyncholaeliocattleya* promoted anthocyanin pigmentation 1 (*RcPAP1*). Although type II R2R3-MYB (Isoform0012108), belonging to subgroup 6, was not found to present a co-expression relationship with ABP structural genes according to WGCNA, its expression profile was consistent with the expression pattern of ABP structural genes at D7 (Figs. 2b and 3b). Therefore, Isoform0012108 was considered a candidate that might activate the ABP and was designated *Rhyncholaeliocattleya* promoted anthocyanin pigmentation 2 (*RcPAP2*). Finally, we confirmed the expression profiles of these candidates using quantitative real-time PCR (qRT-PCR), and the results were consistent with the transcriptome data (Fig. 3b, e).

**Transient overexpression of *RcPCP1* and *RcPAP1/2* in the perianths of a *Phalaenopsis* hybrid.** To verify the function of the putative regulators involved in pigment biosynthesis, *RcPAP1/2* and *RcRCP1* were infiltrated into the perianths of a *Phalaenopsis* hybrid via *Agrobacterium* transformation. The overexpression of the empty vector (EV) did not change the colour of the perianths, which resembled the phenotype of the mock-treated flowers (Fig. 4a). The overexpression of *RcPCP1* or *RcPAP1/2* resulted in white perianths that became slightly yellow or purple-red, respectively (Fig. 4a). We found that the level of carotenoid accumulation was higher in the *RcPCP1*-overexpressing (OE-*RcPCP1*) lines than in the mock- and EV-treated plants and that the transcription levels of *PePSY*, *PeLCYE* and *PeBCH1* were upregulated (Fig. 4b). Notably, *PePSY*s and *BCH1*s remained highly expressed in the hypochile of KOVA at D7 and D8 (Fig. 2c). As expected, the level of anthocyanin accumulation was increased in the perianths of the OE-*RcPAP1/2* lines compared to

the perianths of the mock- and EV-treated plants, and the ABP structural genes from *PeF3H* to *PeANS* were notably upregulated in the OE-*RcPAP1/2* lines (Fig. 4c). These results indicated that *RcRCP1* and *RcPAP1/2* could activate CBP and ABP structural genes and promote yellow or red pigmentation in *Phalaenopsis* flowers, respectively.

**The B-class *AP3*- and *AGL6-like* MADS-box genes participate in flower colour formation.** The B-class *AP3*- and *AGL6-like* MADS-box genes were screened from our transcriptome data, and different subgroups of *AP3* and *AGL6* were identified using phylogenetic trees based on the neighbour-joining (NJ) method (Fig. 5a). The heatmap of these MADS-box genes showed that their expression profiles exhibited spatiotemporal specificity during floral development (Fig. 5a). Among *AP3-like* genes, both the *AP3-1*s and *AP3-4*s were expressed in the perianths and lip. However, the *AP3-1*s were expressed at both D7 and D8, whereas the *AP3-4*s were highly expressed only at D8. Both the *AP3-2*s and *AP3-3*s were expressed at D1 and D4. Spatially, the expression of *AP3-2*s was high in the hypochiles and gradually decreased from the hypochiles to the petals, whereas the expression level of *AP3-3*s did not differ markedly among flower segments. Moreover, the *AGL6-1*s were highly expressed in the perianths at D1 and D4, and their expression level in the sepals was higher than that in the petals (Fig. 5a). Additionally, the *AGL6-2*s were expressed in the lip at all developmental stages, and their expression level in the hypochiles was higher than that in the epichiles. These results showed that different combinations of the *AP3*- and *AGL6-like* MADS-box genes participated in determining the identity of the flower segments during KOVA flower development: *AP3-1/3/4* and *AGL6-1* were associated with the sepals, *AP3-1/2/3/4* and *AGL6-1* were associated with the petals, and *AP3-1/2/3/4* and *AGL6-2* were associated with the lip. Although the *AP3-1/2/3/4*s and *AGL6-2*s were expressed in both lip segments, the transcription level of the *AP3-1/4*s was higher in the purple-red epichile than in the yellow hypochile, whereas the transcription level of *AP3-2*s and *AGL6-2*s in the hypochile was higher than that in the epichile. Although we did not find co-expression relationships of the *AP3*- and *AGL6-like* genes with ABP structural genes according to WGCNA, the *AGL6-2*s were coexpressed with *BCH1*s and *RcPCP1* (Fig. 3c), and the *AP3-1/4*s were coexpressed with the *PSY*s, *ZDS*s and *Z-ISO*s (Fig. 5b) during floral development.

To further investigate the differential expression of *AP3*- and *AGL6-like* genes between the sepal/petal segments, qRT-PCR was used to analyse the expression profiles of these MADS-box genes in the sepal/petal segments at D4 and D7 (Fig. 5c). The results showed that the differences in the expression of *AP3-1/2/4* and *AGL6-2* between sepal/petal segments with similar colours were smaller than those between lip segments with different colours (Figs. 1c and 5c, d). Moreover, the transcriptome data showed that the expression levels of the *AP3-1/2/4*s and *AGL6-2*s differed

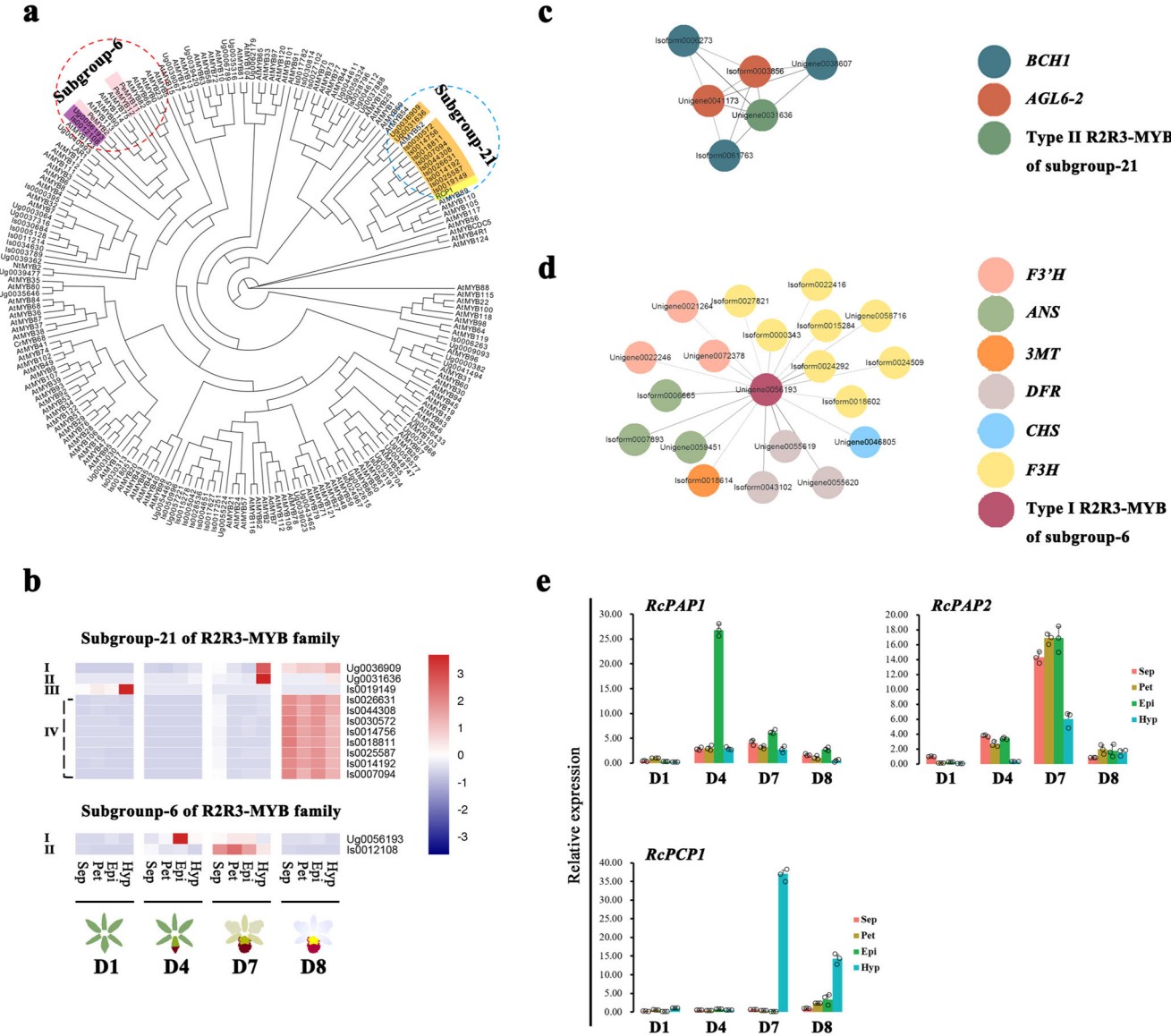

**Fig. 3 The identification of R2R3-MYB candidates involved in pigment biosynthesis using bioinformatics analysis. a** All amino acid sequences of KOVA R2R3-MYBs with pigment-related R2R3-MYBs and all *Arabidopsis* R2R3-MYBs were used to build a phylogenetic tree based on maximum likelihood, which was employed to infer the R2R3-MYB candidates of KOVA involved in pigment biosynthesis. The red and blue circles indicate subgroup 6 and subgroup 21 of the R2R3-MYB family, respectively. Purple and pink indicate the isoforms/unigenes of KOVA and the R2R3-MYBs of *Phalaenopsis*, respectively. Orange and yellow indicate the isoforms/unigenes of KOVA and the R2R3-MYBs of *Mimulus lewisii*, respectively. **b** Heatmap analysis based on the FPKM values of the R2R3-MYB candidates was performed to illustrate the expression profile of these candidates in the perianth and lip segments during flower development. The FPKM value of each candidate is the mean of three biological replicates. **c** The relationship between these candidates and ABP/CBP structural genes was further explored using weighted gene coexpression network analysis (WGCNA). The midnight-blue module shows that the type II R2R3MYB of subgroup 21 is coexpressed with the *BCH1*s and *AGL6-2*s during flower development. The circles represent isoforms/unigenes, and the grey lines represent connectivity among the isoforms/unigenes. **d** The light-yellow module shows that the type I R2R3-MYB of subgroup 6 presents a coexpression relationship with some ABP structural genes. **e** The expression profiles of *RcPAP1*, *RcPAP2* and *RcPCP1* during flower development were confirmed using qRT-PCR. The data are the mean ± SD from three biological replicates. Isoform and unigene are abbreviated as Is and Ug, respectively.

more between the epichile and hypochile than between the sepal and petal (Fig. 5a). These results showed that the expression levels of the *AP3-1/2/4*s and *AGL6-2*s differed more between the epichile and hypochile (the lip segments display obvious phenotypic differences) than between the sepal/petal segments (which show similar phenotypes, especially in terms of colour).

In addition, we identified a peloric mutant, designated KOVA mutant 1 (M1), in which the lip was transformed into petals (Fig. 6a). The colouration of the epichile and hypochile of M1 at D8 changed from the original yellow and purple-red colouration

to very light yellow and almost white coloration, respectively (Fig. 6a). Before full flowering, we could not distinguish M1 from KOVA, and we only collected M1 samples at D8. Therefore, according to the heatmap analysis of transcript expression, we compared the expression profiles of the genes that were expressed at the late developmental stage (i.e., D7 and D8), including *AP3-* and *AGL6*-like genes and genes involved in pigment biosynthesis, between M1 and wild-type flowers at D8 using qRT-PCR. The results showed that the epichile and hypochile displayed a similar phenotype in M1 compared to the wild-type, the differential

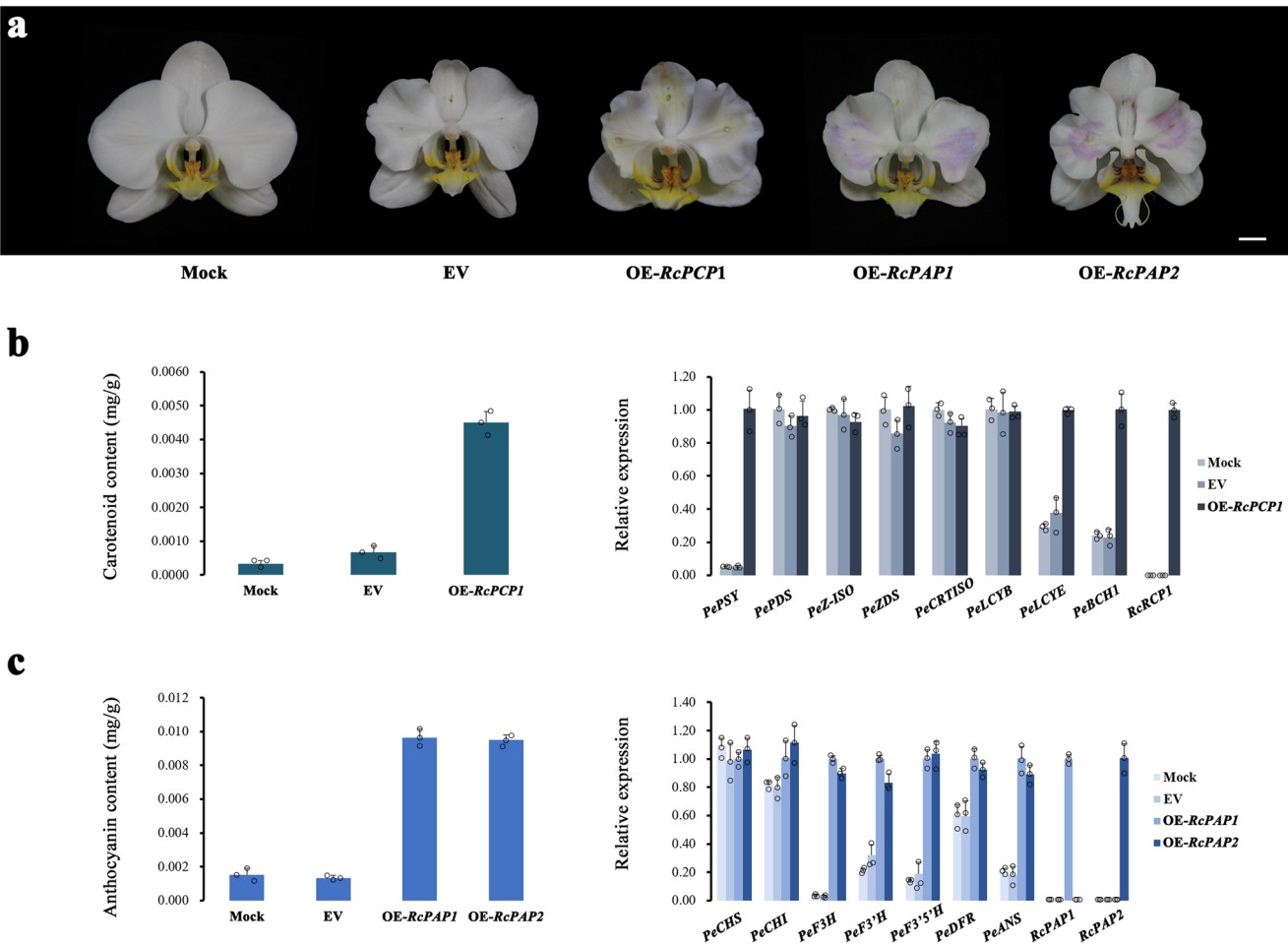

**Fig. 4 Verification of the function of *RcPAP1/2* and *RcPCP1* in *Phalaenopsis* hybrid petals via a transient overexpression assay. a** The white petals of the empty vector-overexpressing (EV) lines were similar to those of the mock-treated plants. *RcPCP1-*, *RcPAP1-*, and *RcPAP2*-overexpressing (OE-*RcPCP1*, OE-*RcPAP1* and OE-*RcPAP2*) lines displayed slightly yellow or red petals. **b** The carotenoid content of the OE-*RcPCP1* petals was higher than that of the mock- and EV-treated petals (left), and the qRT-PCR results showed that the expression levels of some CBP structural genes were increased in OE-*RcPCP1* (right). **c** Anthocyanin accumulation (left) and the expression levels of some ABP structural genes (right) were obviously increased in OE-*RcPAP1/2* petals. The data are the mean ± SD from three biological replicates (Supplementary Fig. 2). Bar = 1 cm.

expression of *AP3-1/4* and *AGL6-2* in the lip segments was lower in M1 than that in the wild-type, and *RcPCP1*, *BCH1* and *RcPAP2* were obviously downregulated in M1 (Fig. 6a, b).

## Discussion
The expression levels of structural genes in pigment biosynthesis pathways can directly influence pigment accumulation to determine flower colour[6,16,17]. However, most studies have focused on either anthocyanins or carotenoids, and little is known about the detailed process of colour differentiation in the sepal and petal segments. Based on a morphological anatomical analysis performed together with large-scale metabolome analysis and transcriptome analysis, our study indicated that colour pattern formation during KOVA floral development could be divided into three phases. In phase I (i.e., D1), the ABP and CBP were inactivated, resulting in an absence of red or yellow flower segments (Figs. 1d and 2b). In phase II (i.e., from D2 to before D7), the whole ABP was strongly activated only in the epichile, which resulted in the substantial accumulation of cyanins and purpled-red colouration of the epichile (Fig. 2a, b). In the final phase (i.e., from D7 to D8), the perianths and hypochile turned pale pink and yellow, respectively. In this phase, the whole ABP was activated in the sepals and petals, and cyanins accumulated.

Nevertheless, the intensity of the activity of the ABP structural genes was lower in the perianths in phase III than that in the epichile in phase II, which might be the reason that the sepals and petals produce only small amounts of cyanins and appear pale pink (Fig. 2a, b). Although the increased lutein content resulted in various degrees of yellow green colouration in the sepals and petals at D7, lutein content significantly decreased at D8 and ultimately resulted in the loss of the yellow colour. According to previous studies, BCH and CYP97A3 can hydroxylate the β-ring of cyclic carotenes to produce the precursor of lutein[16,34,35], and the suppression of the *BCH2* by RNA interference (RNAi) in *Oncidium* 'Gower Ramsey' (GR) flowers causes a reduction in the lutein content[36]. Additionally, CCDs can cleave carotenoids to generate colourless apocarotenoids[17], and the overexpression of *OgCCD1* in GR flowers causes their yellow lip produce white spots[37]. Thus, from D7 to D8, the reduction in the α-carotene and lutein contents of the KOVA perianths and epichile might be caused by the downregulation of *BCH2*s and *CYP97A3*s and the upregulation of *CCD1*s (Fig. 2c). In contrast, the hypochile maintained the highest α-carotene and lutein contents at D8, possibly due to the sustained expression of *BCH1*s and reduced expression of *CCD7*s (Fig. 2c). A decrease in the chlorophyll content resulted in the loss of green colour and a complete shift to yellow colouration in the hypochile (Figs. 1a and 2a). Notably, the

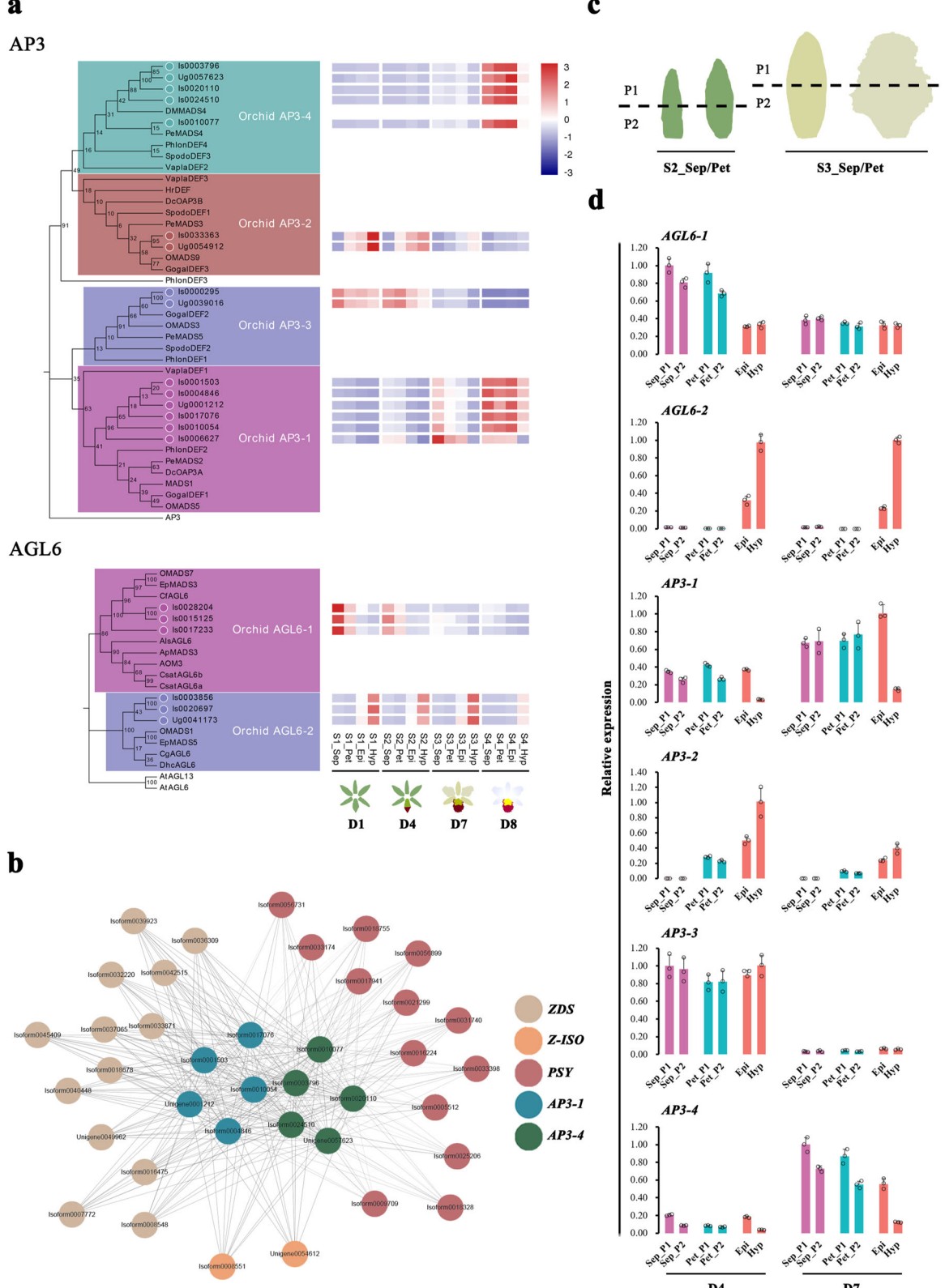

*LCYE*s and *LCYB*s were obviously downregulated in the sepals, petals, and lips at phase III (Fig. 2c). This can be explained by metabolic feedback regulation, which plays an important role in carotenoid accumulation[38]. A previous study showed that the expression level of *LCYE* might present a negative correlation with the lutein content[39]. Therefore, an increased lutein content might result in the downregulation of the expression of *LCYE*s

and *LCYB*s in KOVA flowers. According to the above findings, our study showed that the process of colour differentiation among the floral segments was very complicated. ABP and CBP structural genes were expressed with high spatiotemporal specificity during KOVA flower development, which directly caused the unbalanced distribution of pigments, leading to colour differentiation in the perianths and lip segments.

**Fig. 5 Spatiotemporal expression analysis of the *AP3*- and *AGL6*-like genes in the perianth and lip segments during floral development. a** Phylogenetic trees based on the neighbour-joining method were inferred from the amino acid sequences of KOVA AP3- and AGL6-like MADS-box proteins and those of other plants (left). According to the FPKM values of these KOVA MADS-box genes, their expression profiles in flower segments during floral development were described via heatmap analysis (right). The FPKM value of each isoform/unigene is the mean of three biological replicates. **b** The brown module shows that *AP3-1*s and *AP3-4*s presented a co-expression relationship with some CBP structural genes during floral development according to WGCNA. **c** Sampling strategy for the sepals and petals at D4 (left) and D7 (right), which was similar to the sampling strategy for the lip. These samples were used for qRT-PCR analysis to explore the *AP3*- and *AGL6*-like MADS-box gene expression profiles. **d** The qRT-PCR results showed that the differences in the expression levels of the *AP3*- and *AGL6*-like genes between the perianth segments were smaller than those in the lip segments. The data are the mean ± SD from three biological replicates.

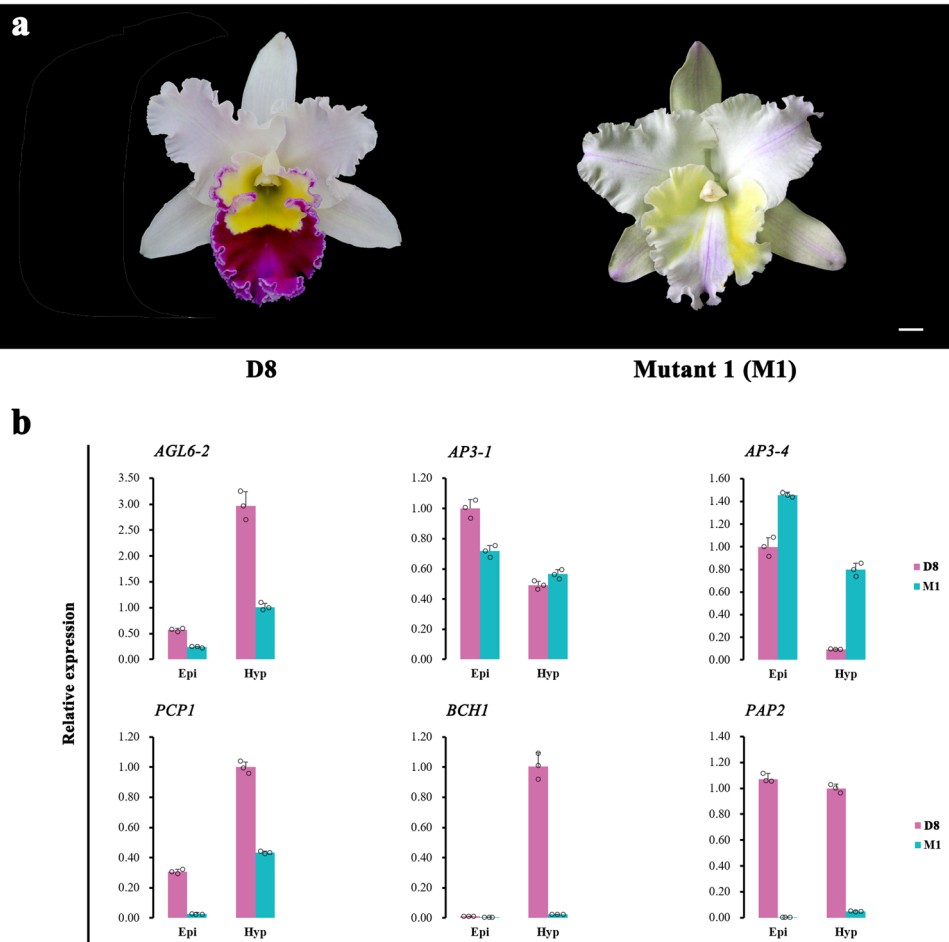

**Fig. 6 The relationship between flower colour and the expression levels of *AP3*- and *AGL6*-like genes. a** Wild-type KOVA flowers at D8 (left) and the KOVA mutant 1 (M1; right). **b** The expression levels of the late-expressed *AP3*- and *AGL6*-like MADS-box genes and some pigment biosynthesis genes were detected using qRT-PCR. The data are the mean ± SD from three biological replicates. Bar = 1 cm.

R2R3-MYBs can regulate the spatiotemporal expression of the structural genes involved in pigment biosynthesis pathways to determine pigmentation intensity and patterning in flowers, which plays a key role in the regulation of flower colour formation[4,40]. In our study, the protein sequences of both *RcPAP1* and *RcPAP2* were found to exhibit high sequence similarity with PeMYB2 (Fig. 3a; Supplementary Fig. 3), which has been shown to promote anthocyanin accumulation in *Phalaenopsis* hybrid flowers[5]. The transient overexpression assay results showed that the overexpression of *RcPAP1/2* activated some ABP structural genes, leading to the accumulation of anthocyanin in *Phalaenopsis* hybrid flowers, which caused the white petals to become purple-red (Fig. 4a, c) and suggested that *RcPAP1/2* are likely to function as an ABP activator in KOVA flowers. Interestingly, *RcPAP1* was highly expressed only in the epichile at D4,

whereas the expression of *RcPAP2* was upregulated in the sepals, petals, and epichile at D7 (Fig. 3b). These results indicated that *RcPAP1* and *RcPAP2* might determine the purple-red colour of the epichile and the pale pink colour of the perianths, respectively, in KOVA. Different regions exist in the amino acid sequences of *RcPAP1* and *RcPAP2* (Supplementary Fig. 3), indicating that they might be derived from different genes with activators that may also be different. Thus, the different activators may have caused the difference in spatiotemporal expression between *RcPAP1* and *RcPAP2*. Additionally, it is worth noting that the activity of ABP structural genes in the epichile at D4 was obviously higher than that in the sepals and petals at D7 (Fig. 2b). However, anthocyanin accumulation and the expression level of ABP structural genes were not obviously different between the OE-*RcPAP1* and OE-*RcPAP2* lines (Fig. 4c). There are two

possible explanations for this finding: R2R3-MYB usually forms an MBW complex with bHLH and WD40 proteins to activate ABP structural genes[41], which indicates that the interacting partner of *RcPAPs* may be downregulated at D7; some MYBs function as repressors and form a complex with bHLH proteins and other activators[41]. It is possible that some repressors were upregulated at D7 to prevent effective MBW complex formation in KOVA flowers. These possibilities need to be tested in the future. In summary, *RcPAP1* and *RcPAP2* might spatiotemporally regulate red colour formation in KOVA flower segments during floral development. The specific expression of the former contributed to the purple-red colour of the epichile at phase II, and the latter was activated in the perianths, resulting in the pale pink coloration of the perianths at phase III. This finding indicates that differences of anthocyanin accumulation among flower segments may be regulated by different R2R3-MYBs during floral development in plants.

RCP1 belongs to subgroup 21 of the R2R3-MYB family and can activate CBP structural genes to increase the carotenoid content in *Mimulus lewisii* flowers[18]. Our study indicated that the protein sequence of *RcPCP1* was homologous to that of RCP1 (Fig. 3a; Supplementary Fig. 4) and that *RcPCP1* was highly expressed only in the hypochile (Fig. 3b, e). Additionally, the WGCNA results suggested that *RcPCP1* presented a co-expression relationship with *BCH1*s during KOVA flower development. Moreover, a slightly yellow colour appeared in the OE-*RcPCP1* petals (Fig. 4a). The carotenoid content and the expression levels of *PePSY*, *PeLCYE*, and *PeBCH1* in the OE-*RcPCP1* petals were higher than those in the mock- and EV-treated petals (Fig. 4b). Notably, the *PSY*s and *BCH1*s maintained higher expression levels in the hypochile of after KOVA flowering (Fig. 2c). These results suggested that the spatiotemporally specific expression of *RcPCP1* might promote α-carotene and lutein accumulation in the hypochile, resulting in the appearance of yellow colouration in phase III. These results also implied that some R2R3-MYBs belonging to subgroup 21 of monocots might possess a similar function to *RCP1* in the regulation of the CBP in flowers. The yellow parts of flowers can help some plants attract pollinators[1,42], and yellow is also one of the most important colours of ornamental flowers. Therefore, *RcPCP1* will be an important target for future studies on the evolution of orchid flower colouration and will be valuable for the breeding of novel ornamental lines or crops.

At present, most studies focus on the mechanism by which the MBW complex regulates flower colour formation, while MADS-box genes, which are dominant actors in the determination of flower organ identity linked to shape and colour, are often ignored. According to the P code of Orchidaceae, the SP complex comprises AP3-1 and AGL6-1 and is necessary for the development of sepals and petals, whereas AP3-2 and AGL6-2 form the determinant unit of the L complex and are exclusively required for lip formation[23]. Our study showed that the *AP3-1/2/3/4*s and *AGL6-2*s were involved in the development of the identity of the lip during KOVA floral development, but the expression levels of *AP3-1/2/4*s and *AGL6-2*s displayed more differences between the yellow hypochile and the purple-red epichile (Fig. 5a), which might be the reason for the obvious phenotypic difference between these two segments. However, the expressional differences in these MADS-box genes between the sepal/petal segments (they show similar phenotypes during floral development, especially regarding coloration) were smaller than the differences compared to the lip segments (Figs. 1c and 5c, d). Moreover, the *AP3-1/4*s and *AGL6-2*s exhibited a co-expression relationship with some CBP structural genes and *RcPCP1* (Figs. 3c and 5b). Additionally, the obvious downregulation of *RcPCP1*, *BCH1*, and *RcPAP2* might reduce the yellow and red coloration of the M1 hypochile and epichile, respectively (Fig. 6b), and the expression

levels of *AP3-1/4* and *ALG6-2* also differed in M1 compared to the wild-type at D8. Within Orchidaceae, the suppression of the expression of *OAGL6-2* in the *Oncidium* Lemon Heart lip using virus-induced gene silencing (VIGS) resulted in the lightening of yellow colouration and the development of green areas on the abaxial lip[23]. In addition, when the E-class MADS-box genes *PeSEP2* and *PeSEP3* were silenced using VIGS in *Phalaenopsis* OX Red Shoes, the sepals and petals showed green colouration and their tips accumulated higher levels of anthocyanins[24]. Therefore, the expression levels of *AP3-* and *AGL6*-like genes in different combinations might contribute to colour differentiation of the perianth and lip segments during floral development in KOVA. In addition, previous studies have shown that changes in the expression levels of MADS-box genes are accompanied by the transformation of flower petal identity, including petal shape and colour[23–30,43], which suggested that MADS-box genes might have multiple functions and be involved in the shape and colour differentiation of sepals and petals; several MADS-box genes have been verified to directly or indirectly regulate CBP[44,45] or ABP[46] structural gene expression to influence colour in fruits, which also supports our finding that MADS-box genes might participate in KOVA flower colour formation. However, how MADS-box genes actually influence flower colour formation is still unknown, and further functional studies in plants involving thorough transgenic systems are needed to illustrate the role of MADS-box genes in this process.

In summary, the *RcPAP1/2*, *RcPCP1* and *AP3/AGL6* MADS-box genes might collectively regulate colour differentiation in perianth and lip segments during KOVA flower development (Fig. 7), which may result in the spatiotemporal expression of ABP and CBP structural genes to determine colour differentiation, as follows: substantial cyanin accumulation in the epichile results in purple-red colouration before flowering; after flowering, the low levels of cyanin and carotenoids in the perianths result in a pale pink colour, and the extremely high α-carotene and lutein contents in the hypochile result in yellow colouration. These findings from this study can enrich our understanding of flower colour pattern formation.

## Methods

**Plant materials and sampling**. *Rhyncholaeliocattleya* Beauty Girl 'KOVA' (KOVA) flowers are very large and exhibit pale pink perianths, a purple-red epichile and a yellow hypochile (Fig. 1a), while KOVA mutant 1 (M1) exhibits a lip that has been transformed into petals (Fig. 6a). The hybrid *Phalaenopsis* flower has white perianths and a light-yellow lip with red stripes (Fig. 4a) and was used for the transient overexpression assay to identify the function of isoforms/unigenes.

All orchids used in this study were grown in a Chinese Academic Forestry greenhouse (Beijing, China) under natural light with daytime and night-time temperatures of 24–30 °C.

We collected sepals, petals, the purple-red region of epichiles and the yellow region of hypochiles from KOVA flowers at D1 (i.e., bub length < 2 cm), D4 (i.e., bub length 4–5 cm), D7 (i.e., two days after flowering), and D8 (i.e., ten days after flowering) (Fig. 1d). The same four tissues of the M1 flowers were also sampled. The areas of transient isoform/unigene overexpression in *Phalaenopsis* hybrid flowers were sampled. All materials were sampled, immediately frozen in liquid nitrogen and then stored at −80 °C.

**Identification of pigments**. According to a previous study[47], flower tissues were weighed (50 mg), ground into a powder in liquid nitrogen and resuspended in 200 μl methanol, which enabled the extraction of flavonoids and carotenoids from the tissue powder. Then, an equal volume of water and dichloromethane was added to the methanol extract, followed by thorough mixing. Finally, the samples were centrifuged at 13,000 rpm for 2 min to separate flavonoids and carotenoids into the supernatant liquid (aqueous) and bottom liquid (non-aqueous).

**Metabolite profiling**. Flavonoids and carotenoids in the sepals, petals, epichiles, and hypochiles of KOVA flowers at D7 and D8 were identified and quantified by using an LC-ESI-MS/MS system (Fig. 1d; HPLC, UFLC SHIMADZU CBM20A system, www.shimadzu.com.cn/; MS, Applied Biosystems 4500 QTrap, www.appliedbiosystems.com.cn/). Moreover, total chlorophyll was extracted and quantified as previously described in ref. [48]. At both stages, three biological replicate

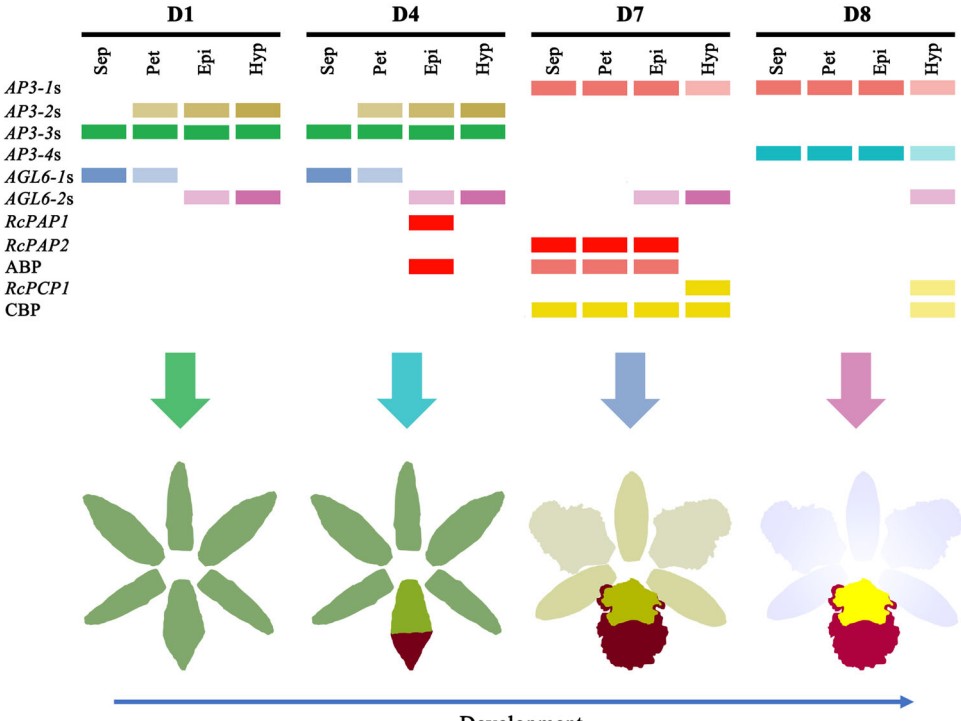

**Fig. 7 The mechanism of colour differentiation in the sepals, petals, and lip segments during KOVA floral development.** Different combinations of *AP3*- and *AGL6*-like MADS-box genes, R2R3-MYBs, and ABP and CBP structural genes might participate in KOVA flower colour pattern formation during floral development. The colour of each rectangle indicates whether the isoforms/unigenes was expressed, and deep colours indicate that the expression intensity of the isoforms/unigenes is high. ABP Anthocyanin biosynthesis pathway; CBP carotenoid biosynthesis pathway.

bundles were evaluated for each part of the flower (i.e., samples of the same segment from four individuals were mixed to form one replicate), for a total of 24 samples. The flavonoids present in the freeze-dried samples were extracted, identified, and quantified using a previously described method[49].

For carotenoid extraction, the freeze-dried samples were ground into powder using a mixer mill (MM 400, Retsch, Germany) with a zirconia bead for 1.5 min at 30 Hz, and 100 mg of the powder was weighed and added to 1 ml of extraction solution 1 (hexane–acetone–ethanol (2:1:1 by volume) containing 0.1% butylated hydroxytoluene). Then, the sample was vortexed for 30 s and centrifuged at 14,000 rpm and 4 °C for 5 min. The supernatant liquid was collected and mixed with 1 ml of the extraction solution, and the mixture was centrifuged again. The final supernatant liquid was dried under a stream of nitrogen gas. A total of 200 µl of extraction solution 2 ((ethyl nitrile-methyl alcohol)-methyl tert-butyl ether (17(3:1):3) by volume) was used to dissolve the dried residues, and the sample was subsequently vortexed for 30 s and centrifuged at 12,000 rpm at 4 °C for 2 min. Finally, the extract residues were filtered through a 0.22-µm membrane filter and stored in brown bottles. Carotenoid quantification in the samples was performed according to a previously described method[50]. In addition, the total carotenoids and anthocyanins of the samples showing transient overexpression were extracted and quantified as previously described[48,51].

**Total RNA extraction and quality assessment.** All freeze-dried samples were ground to a powder in liquid nitrogen and weighed (50 mg) to extract total RNA using an RNA extraction kit (Huayueyang Biotechnology Inc., Beijing, CN) following the specifications of the kit. We used an Agilent 2100 Bioanalyzer (Agilent Technologies Inc., CA, USA) and agarose gel electrophoresis to detect the integrity of RNA, and the purity and concentration of the RNA were analysed by using a NanoDrop™ One/One$^C$ system (Thermo Fisher Scientific, MA, USA).

**PacBio *Iso-Seq* library preparation, sequencing and data analysis.** PacBio *Iso-Seq* library preparation and sequencing were performed as previously described[52] with some modifications. To obtain all full-length transcriptome sequences expressed during KOVA flower development, the total RNA of the sepals, petals, epichiles, and hypochiles from D1, D7 and D8 flowers were fully mixed in equal quantities to construct sequencing libraries. Then, cDNA was synthesised using the SMARTer™ PCR cDNA Synthesis kit. After 14 cycles of PCR amplification, the products with sizes of 1–4 kb and >4 kb size used to construct libraries using the BluePippin™ Size-Selection System (Sage Science, MA, USA). Then, large-scale PCR was performed for the two libraries to amplify full-length cDNA, which was used for SMRTbell library construction (including cDNA damage repair, terminal

repair, and ligation with SMRT dumbbell-type adapters). Before sequencing, the SMRTbell template was annealed with a sequencing primer, and polymerase was linked to the primer-annealed template. The polymerase-bound templates were sequenced on the PacBio Sequel platform using P6-C4 chemistry with 10 h movies.

The unique full-length transcriptome sequences (isoforms) were obtained from the raw sequencing data using the SMRT Link v5.0.1 pipeline[53] supported by Pacific Biosciences. First, the circular consensus sequence (CCS) reads were extracted from the subreads in a BAM file. Second, the CCS reads were classified into full-length non-chimeric (FLNC), non-full-length (nFL), chimeric and short (i.e., read length < 200 bp) reads depending on whether cDNA primers and poly A tails were detected. Subsequently, we used Iterative Clustering for Error Correction (ICE) to cluster the FLNC reads and generate the consensus FLNC reads, which were further polished with nFL reads using Quiver to produce high-quality consensus FLNC reads. Finally, the redundant high-quality consensus FLNC reads were removed using CD-HIT-v4.6.7[54] with a threshold of 0.99 identity to obtain the final isoforms.

The final isoforms were annotated by BLAST searches against the nonredundant protein (Nr) database (http://www.ncbi.nlm.nih.gov) and the Kyoto Encyclopedia of Genes and Genomes (KEGG) database (http://www.genome.jp/kegg) at an E-value threshold of 1e−5. To eliminate isoforms that may have been derived from *Cymbidium mosaic virus* (CymMV), we used CymMV genomes (ref|NC_001812.1|) as a reference to identify virus-originating isoforms according to their similarity (query coverage > 90% and percent identity > 80%) and obtained the final isoform set. The coding sequences (CDSs), protein sequences, and UTR sequences of the isoforms were analysed using ANGEL[55]. We aligned the protein sequences of the isoforms to the Pfam database (version 26.0) by using the Pfam_Scan program[56] and SMART database (version 06/08/2012) with the HMMER program for profile hidden Markov models for biosequence analysis (http://hmmer.org/) to predict the protein domains.

**Illumina transcriptome (RNA-Seq) library preparation, sequencing and expression level estimation.** The total RNA of the sepals, petals, epichiles, and hypochiles from D1, D4, D7, and D8 flower was extracted and used for RNA-Seq sequencing (Fig. 1d) with the aim of describing the transcriptome expression profile during KOVA flower development. In these four stages, three biological replicate bundles from each part of the flower were evaluated (i.e., samples from the same structures of 4–6 individuals were mixed to form one replicate), for a total of 48 samples. *RNA-Seq* library preparation was performed according to a previous study[57]. The final PCR products were sequenced using Illumina HiSeq™ 4000.

We used in-house Perl scripts to process the raw data to obtain the clean data. In this process, reads containing adapters, reads containing more than 10%

unknown nucleotides (N) and low-quality reads containing more than 50% low-quality (Q-value ≤ 10) bases were filtered from the raw data. The rRNA reads were removed from the clean data to obtain the effective data using bowtie2[58]. The effective data were mapped to the CymMV genome (ref|NC_001812.1|) using bowtie2, and the CymMV reads were filtered from the effective data to generate the final clean data. These final clean data were then mapped to the pseudoreference genome (i.e., the previously described final isoform set) using HISAT2[59]. The unmapped reads were assembled de novo with Trinity[60] to obtain the unigene set. The annotation, CDSs, protein sequences, UTR sequences, and protein domain predictions of the unigenes were analysed using a method consistent with the above approach. Then, we combined these unigenes with the PacBio Iso-Seq data, resulting in a final reference transcriptome for KOVA. Isoform and unigene expression levels were estimated via the FPKM method using RSEM[61].

**Phylogenetic analysis and co-expression network construction**. The ML phylogenetic tree was constructed by using RAxML-HPC BlackBox with the default parameters at the CIPRES SCIENCE GATEWAY (http://www.phylo.org/) to analyse the phylogenetic relationships among the R2R3-MYBs of KOVA and the genes from other plants. Sequence alignments were performed with MUSCLE[62]. Then, NJ phylogenetic trees were constructed with bootstrap values estimated from 1000 replicate runs using MEGA7[63] to analyse the phylogenetic relationships among the AP3- and AGL6-like genes of KOVA and the genes from other plants. Phylogenetic trees were modified using Evolgenius (http://www.evolgenius.info/evolview/). Heatmap analysis was used to display the isoform/unigene expression profile using Omiscshare tools (http://omicshare.com/tools/). The WGCNA was performed using the WGCNA (v1.47) package in R[64]. The isoforms/unigenes expression values were imported into the WGCNA package to establish co-expression modules using the automatic network construction function blockwiseModules with deissuesettings, except that the power was 0.5; the TOMType was unsigned; the mergeCutHeigh was 0.8; and the minModuleSize was 50. All of the isoforms/unigenes were finally clustered into 17 modules. The networks were visualised using Cytoscape v3.3.0.

**qRT-PCR analysis**. cDNA was synthesised from total RNA using EasyScript One-Step gDNA Removal and cDNA Synthesis SuperMix (TransGen Biotech, Beijing, China). The primer pairs for each isoform/unigene were designed using NCBI Primer-BLAST (http://www.ncbi.nlm.nih.gov). The cDNA template was fully mixed with 2X TB Green™ Premix Ex Taq™ (TaKaRa, Japan) and used for qRT-PCR in a LightCycler 480 System (Roche, USA). The qRT-PCR protocol involved an initial denaturation step (95 °C for 30 s, ramp rate of 4.4 °C/s), followed by 45 cycles of PCR (95 °C for 5 s, ramp rate of 4.4 °C/s; 60 °C for 30 s, ramp rate of 2.2 °C/s, acquisition mode: single), melting (analysis mode: melting curve, 95 °C for 5 s, ramp rate of 4.4 °C/s; 60 °C for 1 min, ramp rate of 2.2 °C/s; 95 °C, acquisition mode: continuous, acquisitions: 5/°C), and cooling (50 °C for 30 s, ramp rate of 2.2 °C/s). From D1 to D8, the transcriptional levels of the isoforms/unigenes were calculated from three biological samples. The 18S housekeeping gene of KOVA and PeActin4[65] were used for normalisation. All PCR primers are listed in the supplementary materials (Supplementary Table 1).

**Transient overexpression of the R2R3-MYBs by Agrobacterium infiltration**. The CDSs of RcPCP1, RcPAP1 and RcPAP2 were cloned by using the corresponding cloning primers, which were designed based on transcriptome data (Supplementary Table 1). Overexpression vector construction and infiltration were performed using a previously described method[5] with some modifications. Briefly, RcRCP1, RcPA1 and RcPA2 were ligated into pCAMBIA1304 by using T4 ligase. The recombinant plasmids were transformed into Agrobacterium tumefaciens EHA105 via heat activation. After incubation, the EHA105 solution containing the overexpression vectors was centrifuged, and bacterial cell pellets were then resuspended by adding liquid infection medium (Murashige and Skoog liquid medium containing 1 mM MES and 100 μM acetosyringone) to an OD$_{600}$ = 1. The final suspensions were injected into the petals of the Phalaenopsis hybrid two days before flowering by using an injector. Subsequently, the infected Phalaenopsis hybrid was cultivated in a Chinese Academic Forestry greenhouse under natural light with daytime and night-time temperatures ranging from 24 to 30 °C for 6−8 days. After the emergence of a stable phenotype, the area of transient isoform/unigene overexpression in the flower was photographed, and RNA was extracted. To ensure the reliability of the results, the phenotypes assessment in the transient overexpression assay of each isoform/unigene was independently repeated at least three times.

**Statistics and reproducibility**. Significant test was conducted by Student's paired t-test using Omicsshare tools (http://omicshare.com/tools/), and asterisk and double asterisk indicate values that differ significantly from those in the sample with the highest value at $P < 0.05$ and $P < 0.01$, respectively. In the experiment of RNA-Seq sequencing and metabolite profiling, three biological replicate bundles from each part of the flower were evaluated (i.e., samples from the same structures of 4–6 individuals were mixed to form one replicate). For the transient overexpression assays, the phenotypes assessment in the transient overexpression assay of each isoform/unigene was independently repeated at least three times to ensure

the reliability and reproducibility of the results. Additionally, all of the qRT-PCR experiments in our study were performed three biological replications and the transcriptional levels of the isoforms/unigenes were calculated from three biological samples.

**Reporting summary**. Further information on research design is available in the Nature Research Reporting Summary linked to this article.

## Data availability

The RNA-Seq data have been deposited in NCBI under SRA accession codes: PRJNA559603. The Iso-Seq data have been deposited in NCBI under SRA accession codes: PRJNA559608. The sequence of RcPAP1, RcPAP2 and RcPCP1 have been deposited in NCBI under GenBank accession numbers: BankIt2261106 RcPCP1 MN420461, BankIt2261154 RcPAP2 MN420462 and BankIt2261158 RcPCP1 MN420463.

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

## Acknowledgements

We are grateful to Dr. Li-Wei Chu (Chinese Academy of Forestry) for help with experiments. This work was funded by "The import of important parents of *Cattleya* and targeted breeding technology of lip" (Grant No. 2014-4-15).

## Author contributions

Y.W. managed the project; Y.W., Z-J.L. and B-J.L. designed experiment and coordinated the project; Y.W., Z-J.L. and B-J.L. wrote the paper; J-Y.W., W-C.T. and D.-Y.Z. reviewed the paper, B-J.L. and B-Q.Z. grew the plant material; B-J. L., B-Q. Z., L-H. Z. and X. W. collected and prepared samples; J-Y. W. assisted B.-J. L. in analysing transcriptome and metabolome data; B-J. L. participated in all the experiments; B-Q. Z., W-C. T. and H.-C. L. assisted in completing transient overexpression assay; H-J. Q. assisted in measuring total chlorophyll content of samples.

## Competing interests

The authors declare no competing interests.
