## [Peer Review File · Communications Biology]

Reviewers' comments:

Reviewer #1 (Remarks to the Author):

In this study Li et al. carried out a series of transcriptome analyses and identified two subgroups of R2R3-MYB genes involved in spatial patterning of flower coloration in the ornamentally important *Cattleya* hybrid orchid. One subgroup of MYB regulates carotenoid pigmentation in the hypochile of the lip, and the other regulates anthocyanin pigmentation in the epichile. In addition, they identified some AP3- and AGL6-like MADS-box genes that may be upstream of the MYB genes. The manuscript is easy to read in terms of logic flow, and the figures are beautiful. I find two aspects of the work particularly interesting.

(1) A homolog of RCP1, a subgroup-21 MYB that was previously reported to regulate carotenoid biosynthesis in monkeyflowers, seems to also regulate carotenoid pigmentation in orchids. Transient assay using 35S:RcPCP1 construct in *Phalaenopsis* petals resulted in light yellow color and up-regulation of some carotenoid biosynthesis genes. A recent review (<https://doi.org/10.3389/fpls.2019.01017>) has lamented over the fact that lots of papers have reported putative carotenoid-activating transcription factors in various plants but none of these factors seems to be conserved across different plant systems. The 35S:RcPCP1 transient assay results are noteworthy, because they suggest that at least one gene (i.e., RCP1-like R2R3-MYB) can activate carotenoid biosynthesis in both monocots and eudicots. Because I think this is probably the most significant result, I suggest the authors to consider the following to enhance this aspect of the paper: (i) show the protein sequence alignment of RcPCP1, RCP1, and the *Arabidopsis* subgroup-21 MYBs in a supplementary figure, highlighting the "signature motif" that defines subgroup-21 MYBs; (ii) repeat the transient assay at least one more time (if it hasn't been repeated before), and show more images in the supplementary materials, to make sure this result is reproducible; and (iii) if possible, optimize the experimental conditions for the 35S:RcPCP1 transient assay (e.g., different age of the petal tissue used for the assay; different *Agrobacterium* OD, etc.), because the current results are not so strong: the yellow is so light, in contrast to the much more conspicuous anthocyanin pigmentation in the 35S:RcPAP1 assay. With that said, I realize that the authors might have already done their best to optimize the experimental conditions. If that's the case, please ignore the third suggestion.

(2) The findings that different combinations of AP3- and AGL6-like MADS-box genes may act upstream of the R2R3-MYB or/and the ABP/CBP structural genes are also potentially significant. However, this part of the results is relatively weak. Did you try the transient assay with AP3- and AGL6-like genes in *Phalaenopsis* petals? If you did, what are the results? If not, this transient assay is worth a try. Because the evidence supporting the connection between the MADS-box genes and the R2R3-MYB or ABP/CBP genes are weak, I am not sure Figure 7b makes that much sense.

Minor comments:

191-192. What are the BCH genes described in this paper? Based on studies in *Arabidopsis*, the CYP97s are the carotenoid beta-hydroxylase on the alpha-carotene pathway leading to lutein, and the BCHs are the carotenoid beta-hydroxylase on the beta-carotene pathway leading to zeaxanthin, violaxanthin, and neoxanthin. Given that the bulk of the carotenoids in this orchid is lutein, shouldn't we focus on the CYP97s instead of the BCHs?

215-217, RcPCPs should be corresponding to CBP and RcPAPs to ABP. The order is the opposite as written.

266. Please spell out the full name of "P. hybrids"

411. What is the "P. orchid flowers"?

434-436. This statement doesn't seem right. It has been shown in multiple systems that different R2R3-MYBs regulate anthocyanin accumulation in different flower parts (e.g., ROSEA vs. Venosa in snapdragon, AN2 vs. DPL in putunia, PLEAN vs. NEGAN in monkeyflower)

Figure 4. It should be noted in the figure legend that the transient assay was performed in *Phalaenopsis* petals.

Reviewer #2 (Remarks to the Author):

Li et al. investigated the color patterning process during flower development, identified pigment components of different segments of the perianth, and uncovered the expression profiles of structural genes involved in the pigment biosynthesis pathways throughout RNA-seq, by using *Rhyncholaelio cattleya* Beauty Girl 'KOVA' as a model. In particular, they found that the spatiotemporal specificity of pigment accumulation is differentially regulated by three R2R3 MYB genes by expression and functional studies. They further identified the expression patterns of AP3- and AGL6-like genes in KOVA and tried to reveal the regulatory relationship between the pigment biosynthesis genes and floral MADS-box genes by comparing the expression levels of these genes in wild type and a peloric mutant flowers. The results are interesting, but the manuscript was not well organized.

In particular, the authors spent a whole section to describe the expression patterns of AP3- and AGL6-like genes in wild type and a peloric mutant flowers and tried to establish a link between floral MADS-box genes and pigment biosynthesis genes. However, the current data seems to be insufficient for the conclusion "the different combinations of AP3/AGL6 MADS-box genes as upstream regulatory factors regulate the expression of structural genes in the pigment biosynthesis pathways or R2R3-MYBs to directly or indirectly influence color differentiation among flower segments." Similarly, the model shown in Fig. 7b did not reflect their idea.

Specific comments:

1. Line 35, "... was directly regulated by three R2R3-MYB transcription factors" – The authors did not show evidence supporting the direct regulation of the three R2R3-MYB transcription factors to the pigment biosynthesis genes.
2. Two recent studies have revealed the mechanisms underlying the color pattern of *Antirrhinum majus pseudomajus*, which should be consulted in the Introduction:
 - a) Bradley, D. et al. Evolution of flower color pattern through selection on regulatory small RNAs. *Science* 358, 925–928 (2017).
 - b) Tavares, H. et al. Selection and gene flow shape genomic islands that control floral guides. *Proc. Natl. Acad. Sci. USA* 115, 11006–11011 (2018).
3. Lines 120-123, to avoid confusion, I suggest that the authors use D2, D4, D7 and D8 to directly name the samples, instead of S1, S2, S3 and S4.
4. Lines 127-129, "The total carotenoid content of the hypochile was significantly higher than that of other flower segments at both stages (Fig. 2a)." – The total carotenoid content is also higher in the S3 sepals than most samples. The statistical analysis is required for the significance test.
5. Lines 214-217, the conclusion "eleven *Rhyncholaelio cattleya* Promoted Carotenoid Pigmentations (RcPCPs) and two *Rhyncholaelio cattleya* Promoted Anthocyanin Pigmentations (RcPAPs) were likely to activate ABP and CBP (which should be CBP and ABP) structural genes, respectively." cannot be drawn only based on the phylogenetic tree.

6. Lines 258-259, "Type I RcPAP (RcPAP1) and type II RcPAP (Isoform0012108, RcPAP2) are candidates that may promote the expression of ABP structural genes." – How did the authors draw the conclusion of type II RcPAP2 gene? I did not find any evidence showing the relationship between type II RcPAP and ABP structural genes.
7. Lines 311-313, the expression pattern of AP3-2 is against the statement "The transcriptional level of the AP3-1/2/4s in the purple-red epichile was higher than that in the yellow hypochile, whereas the transcriptional level of AGL6-2s in the hypochile was higher than that in the epichile."
8. Lines 333-335, "Interestingly, the results showed that the expression levels of AP3- and AGL6-like genes were not significantly altered between the sepal and petal segments with similar colors compared to lip (Fig. 5c)." – The conclusion did not make sense. It would be better to compare the qRT-PCR results with the expression patterns as revealed by RNA-seq data and draw a conclusion of the express of AP3- and AGL6-like genes by integrating the two lines of evidence.
9. Lines 340-343, please provide reasons why the authors selected AGL6-2, AP3-1, AP3-4, PCP1, BCH1 and PAP2 as representative genes?
10. Lines 348-350, "Additionally, WGCNA results showed that the AGL6-2s were co-expressed with BCH1s and RcPCP1 (Fig. 3c), while AP3-1/4s were co-expressed with PSYs, ZDSs, and Z-ISOs (Fig. 6c) during floral development." – How do the authors interpret this finding given that AP3-1/4 and AGL6 show differential expression in the purple-red epichile where ABP genes are dominant and yellow hypochile where CBP genes are dominant? Are there any AGL6- or AP3-like genes coexpressed with ABP genes?
11. Lines 350-351, "These results indicated that changes in the expression levels of the AP3- and AGL6-like MADS-box genes might influence the expression of genes involved in pigmentation." – Please be more specific about the function of the AP3- and AGL6-like genes in regulating ABP and CBP genes as well as their contribution to color patterns in the epichile and hypochile.
12. Lines 419-421, "Specific amino acid sequences were present in RcPAP1 and RcPAP2 (Supplementary Fig. 2), indicating that they were derived from different genes with activators that may also be different." – What are specific sequences in RcPAP1 and RcPAP2? In addition, in Supplementary Fig. 2, the authors just provided the amino acid sequences, rather than alignment, of RcPAP1 and RcPAP2 (not PcPA1 and PcPA2).
13. Lines 647-648, "ML and NJ methods were used to analyze the phylogenetic relationships among isoforms/unigenes and genes from other plants, respectively." – However, only a ML tree was reconstructed for R2R3 MYB genes, and a NJ tree was reconstructed for floral MADS-box gens.
14. According to the description (Lines 397-398) in the Discussion, "... the expression level of LCYE might have a negative correlation with the lutein content." However, all the regulatory relationships, including promotion and repression, were represented by arrows in Fig. 2c, which may make your readers confused. In addition, BCH1 and BCH2 as mentioned in the text should be reflected in Fig. 2c.
15. In Fig. 3a, what did different colors indicate in the phylogenetic tree?
16. What did "PeRcRCP1" in Fig 4b and "RcPA1 and PcPA2" in Fig. 4c mean?
17. In Fig. 5, the phylogenetic trees with bootstrap values should be provided for the AP3 and AGL6 trees. In addition, the Orchid AP3-4 lineage is not monophyletic if the current tree is reliable.
18. In Fig. 6b, delete "Rc" from the gene names of RcPCP1 and RcPAP2.

Reviewer #3 (Remarks to the Author):

Li et al. investigated the mechanism of floral colour differentiation among perianth and lip segments in orchids. Based on systematic characterization of genes involved in pigment biosynthesis pathways, they identified and validated by experiment the key transcription factors regulating flower color differentiation. The unbalanced pigment distribution among flower segments is of interesting biological significance, and their results provide an insightful clue to the relevant field. The manuscript is

carefully written.

major

It would be interesting to show more result or discussion about the evolutionary feature of these transcription factors, as well as their potential role in regulating pigment distribution in other flowering plants.

Response to the reviewers' comments

Reviewer 1:

General comment: In this study Li et al. carried out a series of transcriptome analyses and identified two subgroups of R2R3-MYB genes involved in spatial patterning of flower coloration in the ornamentally important *Cattleya* hybrid orchid. One subgroup of MYB regulates carotenoid pigmentation in the hypochile of the lip, and the other regulates anthocyanin pigmentation in the epichile. In addition, they identified some AP3- and AGL6-like MADS-box genes that may be upstream of the MYB genes. The manuscript is easy to read in terms of logic flow, and the figures are beautiful. I find two aspects of the work particularly interesting.

Response: Thank you for your patience and suggestions. We have improved the manuscript. The major revised portions are indicated in red in this revised manuscript.

Comment 1: Show the protein sequence alignment of RcPCP1, RCP1, and the subgroup-21 MYBs in a supplementary figure, highlighting the "signature motif" that defines subgroup-21 MYBs.

Response: Thank you very much. We have provided alignments of RcPCP1, RCP1, and the *Arabidopsis* subgroup 21 MYBs in a supplementary figure and defined the signature motif as "FxDEL" (please see Supplementary Fig. 3)

Comment 2: Repeat the transient assay at least one more time (if it hasn't been repeated before), and show more images in the supplementary materials, to make sure this result is reproducible.

Response: Thank you very much. We have provided the transient overexpression replicates of RcPCP1, RcPAP1 and RcPAP2 in a supplementary figure to verify that our results are reproducible. (please see supplementary Fig. 2)

Comment 3: If possible, optimize the experimental conditions for the 35S:RcPCP1 transient assay (e.g., different age of the petal tissue used for the assay; different

agrobacterium OD, etc.), because the current results are not so strong: the yellow is so light, in contrast to the much more conspicuous anthocyanin pigmentation in the 35S:RcPAP1 assay. With that said, I realize that the authors might have already done their best to optimize the experimental conditions. If that's the case, please ignore the third suggestion.

Response: Thank you very much. We tested the transient overexpression assays under many different experimental conditions, and the experimental conditions described in the manuscript were the most effective.

Comment 4: The findings that different combinations of AP3- and AGL6-like MADS-box genes may act upstream of the R2R3-MYB or/and the ABP/CBP structural genes are also potentially significant. However, this part of the results is relatively weak. Did you try the transient assay with AP3- and AGL6-like genes in *Phalaenopsis* petals? If you did, what are the results? If not, this transient assay is worth a try. Because the evidence supporting the connection between the MADS-box genes and the R2R3-MYB or ABP/CBP genes are weak, I am not sure Figure 7b makes that much sense.

Response: Thank you very much.

(i) We transiently overexpressed *AP3-1* and *AGL6-2* in *Phalaenopsis* hybrid flowers, but the colour phenotype in the area of injection was similar to that in the mock and EV lines, and the total carotenoid content was not significantly different between the overexpression, mock and EV lines (please see the image below this response). The reason for this lack of differences might be that when we performed the transient overexpression assay, the sepals and petals had already been formed, and the MADS-box genes related to perianth formation had been expressed or were expressed at that time, causing our assay to not work effectively. (ii) The lack of a stable transgenic system in orchids prevents us from further verifying the roles of these MADS-box genes in orchids. However, previous studies have shown that changes in the expression levels of MADS-box genes are accompanied by the transformation of flower petal identity, including flower shape and colour, which suggests that

MADS-box genes might have multiple functions and be involved in the shape and colour differentiation of sepals and petals. Additionally, several MADS-box genes have been verified to directly or indirectly regulate CBP or ABP structural gene expression to influence colour in fruits, which also supports our findings. Therefore, we have changed the description in the manuscript to indicate that we only found a co-expression relationship between MADS-box genes and genes involved in pigment biosynthesis, which indicated that the MADS-box genes might participate in flower colour differentiation during KOVA floral development and provides a new view regarding flower colour differentiation to the scientific community (please see revised manuscript lines 493-530).

Response Figure. Verification of the function of *AP3-1* and *AGL6-2* in *Phalaenopsis* hybrid petals via a transient overexpression assay. The white petals of the empty vector-, *AP3-1*- and *AGL6-2*-overexpressing lines (EV, OE-*AP3-1*, OE-*AGL6-2*) were similar to those of the mock-treated flowers. The expression levels of CBP structural genes were similar among these lines.

Comment 5: 191-192. What are the BCH genes described in this paper? Based on studies in *Arabidopsis*, the CYP97s are the carotenoid beta-hydroxylase on the alpha-carotene pathway leading to lutein, and the BCHs are the carotenoid beta-hydroxylase on the beta-carotene pathway leading to zeaxanthin, violaxanthin, and neoxanthin. Given that the bulk of the carotenoids in this orchid is lutein, shouldn't we focus on the CYP97s instead of the BCHs?

Response: Thank you for your suggestion. Both BCH and CYP97A3 can hydroxylate the β -ring of cyclic carotenes to produce the precursor of lutein, which has been found

in some plants; please refer to Zhu et al.(10.1016/j.abb.2010.07.028), Colasuonno et al. (10.1186/s12864-016-3395-6), and Wang et al.(10.1111/plb.12399). Additionally, according to your suggestion, we screened homologous CYP97A3s from our transcriptome data, which have been shown and discussed in our article. (please see the revised manuscript, lines 198-200; 416-419; Fig. 2c)

Comment 6: 215-217.RcPCPs should be corresponding to CBP and RcPAPs to ABP. The order is the opposite as written.

Response: Thank you very much. We have corrected this issue in our article.

Comment 7: 266. Please spell out the full name of "P. hybrids"

Response: Thank you very much. We have corrected this issue, and "P. hybrids" has been changed to "*Phalaenopsis* hybrid" in our article.

Comment 8: 411. What is the "P. orchid flowers"?

Response: Thank you very much. We have corrected this issue, and "P. hybrids" has been changed to "*Phalaenopsis* hybrid" in our article.

Comment 9: 434-436. This statement doesn't seem right. It has been shown in multiple systems that different R2R3-MYBs regulate anthocyanin accumulation in different flower parts (e.g., ROSEA vs. Venosa in snapdragon, AN2 vs. DPL in putunia, PLEAN vs. NEGAN in monkeyflower)

Response: Thank you very much. We have deleted this sentence in our article.

Comment 10: Figure 4. It should be noted in the figure legend that the transient assay was performed in *Phalaenopsis* petals.

Response: Thank you very much. We have addressed this problem in Figure 4 (please see the revised manuscript, Figure 4).

Reviewer 2:

General comment: Li et al. investigated the color patterning process during flower development, identified pigment components of different segments of the perianth, and uncovered the expression profiles of structural genes involved in the pigment biosynthesis pathways throughout RNA-seq, by using *Rhynchoaelio cattleya* Beauty Girl ‘KOVA’ as a model. In particular, they found that the spatiotemporal specificity of pigment accumulation is differentially regulated by three R2R3 MYB genes by expression and functional studies. They further identified the expression patterns of AP3- and AGL6-like genes in KOVA and tried to reveal the regulatory relationship between the pigment biosynthesis genes and floral MADS-box genes by comparing the expression levels of these genes in wild type and a peloric mutant flowers. The results are interesting, but the manuscript was not well organized.

In particular, the authors spent a whole section to describe the expression patterns of AP3- and AGL6-like genes in wild type and a peloric mutant flowers and tried to establish a link between floral MADS-box genes and pigment biosynthesis genes. However, the current data seems to be insufficient for the conclusion “the different combinations of AP3/AGL6 MADS-box genes as upstream regulatory factors regulate the expression of structural genes in the pigment biosynthesis pathways or R2R3-MYBs to directly or indirectly influence color differentiation among flower segments.” Similarly, the model shown in Fig. 7b did not reflect their idea.

Response: Thank you for your patience and suggestions. We have improved the manuscript. The major revised portions of the text are indicated in red in this version of the manuscript.

Comment 1: Line 35, “... was directly regulated by three R2R3-MYB transcription factors” – The authors did not show evidence supporting the direct regulation of the three R2R3-MYB transcription factors to the pigment biosynthesis genes.

Response: Thank you very much. We have revised this inappropriate expression (please see revised manuscript, lines 35-37).

Comment 2: Two recent studies have revealed the mechanisms underlying the color pattern of *Antirrhinum majus pseudomajus*, which should be consulted in the Introduction:

a) Bradley, D. et al. Evolution of flower color pattern through selection on regulatory small RNAs. *Science* 358, 925–928 (2017).

b) Tavares, H. et al. Selection and gene flow shape genomic islands that control floral guides. *Proc. Natl. Acad. Sci. USA* 115, 11006–11011 (2018).

Response: Thank you very much. These references have been cited in the Introduction of our article.

Comment 3: Lines 120-123, to avoid confusion, I suggest that the authors use D2, D4, D7 and D8 to directly name the samples, instead of S1, S2, S3 and S4.

Response: Thank you very much. S1, S2, S3 and S4 have been changed to D1, D4, D7 and D8 in our article.

Comment 4: Lines 127-129, “The total carotenoid content of the hypochile was significantly higher than that of other flower segments at both stages (Fig. 2a).” – The total carotenoid content is also higher in the S3 sepals than most samples. The statistical analysis is required for the significance test.

Response: Thank you very much. We have corrected this sentence and performed the significance test (please see revised manuscript, lines 127-128; Fig. 2a).

Comment 5: Lines 214-217, the conclusion “eleven *Rhyncholaelio cattleya* Promoted Carotenoid Pigmentations (RcPCPs) and two *Rhyncholaelio cattleya* Promoted Anthocyanin Pigmentations (RcPAPs) were likely to activate ABP and CBP (which should be CBP and ABP) structural genes, respectively.” cannot be drawn only based on the phylogenetic tree.

Response: Thank you very much. We have corrected these descriptions in our article (please see the revised manuscript, lines 219-223).

Comment 6: Lines 258-259, “Type I RcPAP (RcPAP1) and type II RcPAP (Isoform0012108, RcPAP2) are candidates that may promote the expression of ABP structural genes.” – How did the authors draw the conclusion of type II RcPAP2 gene? I did not find any evidence showing the relationship between type II RcPAP and ABP structural genes.

Response: Thank you very much. We have added the reason that Isoform0012108 is considered a candidate that may stimulate ABP structural genes in our article (please see the revised manuscript, lines 280-285).

Comment 7: Lines 311-313, the expression pattern of AP3-2 is against the statement “The transcriptional level of the AP3-1/2/4s in the purple-red epichile was higher than that in the yellow hypochile, whereas the transcriptional level of AGL6-2s in the hypochile was higher than that in the epichile.”

Response: Thank you very much. We have corrected this issue (please see the revised manuscript, lines 339-342).

Comment 8: Lines 333-335, “Interestingly, the results showed that the expression levels of AP3- and AGL6-like genes were not significantly altered between the sepal and petal segments with similar colors compared to lip (Fig. 5c).” – The conclusion did not make sense. It would be better to compare the qRT-PCR results with the expression patterns as revealed by RNA-seq data and draw a conclusion of the express of AP3- and AGL6-like genes by integrating the two lines of evidence.

Response: Thank you very much. We agree with you and have corrected this part of the manuscript based on your suggestions (please see revised manuscript, lines 369-375).

Comment 9: Lines 340-343, please provide reasons why the authors selected AGL6-2, AP3-1, AP3-4, PCP1, BCH1 and PAP2 as representative genes?

Response: Thank you very much. We have corrected this part of the manuscript based

on your suggestions (please see the revised manuscript, lines 379-385).

Comment 10: Lines 348-350, “Additionally, WGCNA results showed that the AGL6-2s were co-expressed with BCH1s and RcPCP1 (Fig. 3c), while AP3-1/4s were co-expressed with PSYs, ZDSs, and Z-ISOs (Fig. 6c) during floral development.” – How do the authors interpret this finding given that AP3-1/4 and AGL6 show differential expression in the purple-red epichile where ABP genes are dominant and yellow hypochile where CBP genes are dominant? Are there any AGL6- or AP3-like genes coexpressed with ABP genes?

Response: Thank you very much. According to the P code of Orchidaceae, the SP complex comprises AP3-1 and AGL6-1 and is necessary for the formation of sepals and petals, whereas AP3-2 and AGL6-2 form the determinant unit of the L complex and are exclusively required for lip formation (Hsu, H. F. et al. Model for perianth formation in orchids. *Nat. Plants* 1, 15064 (2015)). The different combinations of AP3- and AGL6-like MADS-box genes contribute to the formation of the identity of different flower segments. The purple-red epichile and yellow hypochile are components of the lip in KOVA, and *AP3-1/2/3/4* and *AGL6-2* are expressed in both of these structures during floral development. However, a significant phenotypic difference was observed between the purple-red epichile and yellow hypochile. Therefore, the different expression levels of the *AP3-1/2/4s* and *AGL6-2s* between the epichile and hypochile might result in phenotypic differences in both lip segments (please see revised manuscript, lines 342-346; 381-389; 494-531).

Comment 11: Lines 350-351, “These results indicated that changes in the expression levels of the AP3- and AGL6-like MADS-box genes might influence the expression of genes involved in pigmentation.” – Please be more specific about the function of the AP3- and AGL6-like genes in regulating ABP and CBP genes as well as their contribution to color patterns in the epichile and hypochile.

Response: Thank you very much. We have deleted this sentence and revised the description in this section and the discussion about MADS-box genes (please see

revised manuscript, lines 381-389; 494-531).

Comment 12: Lines 419-421, “Specific amino acid sequences were present in RcPAP1 and RcPAP2 (Supplementary Fig. 2), indicating that they were derived from different genes with activators that may also be different.” – What are specific sequences in RcPAP1 and RcPAP2? In addition, in Supplementary Fig. 2, the authors just provided the amino acid sequences, rather than alignment, of RcPAP1 and RcPAP2 (not PcPA1 and PcPA2).

Response: Thank you very much. We have added the alignment of the amino acid sequences of RcPAP1, RcPAP2 and PeMYB2, which has been provided in our supplementary materials. Additionally, we have revised the description in our article (please see the revised manuscript, lines 454-458, Supplementary Fig. 4).

Comment 13: Lines 647-648, “ML and NJ methods were used to analyze the phylogenetic relationships among isoforms/unigenes and genes from other plants, respectively.” – However, only a ML tree was reconstructed for R2R3 MYB genes, and a NJ tree was reconstructed for floral MADS-box genes.

Response: Thank you very much. We have corrected this description in our article (please see the revised manuscript, lines 671-677).

Comment 14: According to the description (Lines 397-398) in the Discussion, “... the expression level of LCYE might have a negative correlation with the lutein content.” However, all the regulatory relationships, including promotion and repression, were represented by arrows in Fig. 2c, which may make your readers confused. In addition, BCH1 and BCH2 as mentioned in the text should be reflected in Fig. 2c.

Response: Thank you very much. We have modified Fig. 2c in our article (please see revised manuscript, Figure 2).

Comment 15: In Fig. 3a, what did different colors indicate in the phylogenetic tree?

Response: Thank you very much. We have added a description of the meaning of the different colours in Fig. 3a. (please see the revised manuscript, Fig. 3)

Comment 16: What did “PeRcRCP1” in Fig 4b and “RcPA1 and PcPA2” in Fig. 4c mean?

Response: Thank you very much. We have corrected this issue in Fig. 4 b, c. (please see the revised manuscript, Figure 4)

Comment 17: In Fig. 5, the phylogenetic trees with bootstrap values should be provided for the AP3 and AGL6 trees. In addition, the Orchid AP3-4 lineage is not monophyletic if the current tree is reliable.

Response: Thank you very much. We have added bootstrap values in Fig. 5 (please see revised manuscript, Figure 5).

Comment 18: In Fig. 6b, delete “Rc” from the gene names of RcPCP1 and RcPAP2.

Response: Thank you very much. We have corrected this issue in Fig. 6b (please see revised manuscript, Figure 6).

Reviewer 2:

General comment: Li et al. investigated the mechanism of floral colour differentiation among perianth and lip segments in orchids. Based on systematic characterization of genes involved in pigment biosynthesis pathways, they identified and validated by experiment the key transcription factors regulating flower color differentiation. The unbalanced pigment distribution among flower segments is of interesting biological significance, and their results provide an insightful clue to the relevant field. The manuscript is carefully written.

Response: Thank you for your patience and suggestions. We have improved the manuscript. The major revised portions of the text are indicated in red in this version of the manuscript.

Comment 1: It would be interesting to show more result or discussion about the evolutionary feature of these transcription factors, as well as their potential role in regulating pigment distribution in other flowering plants.

Response: Thank you very much for your valuable suggestions. We have added the alignment of these transcription factors, including RcPAP1, RcPAP2, and RcPCP1, with their closely related genes from other plants. Additionally, we have added a discussion of the role of MADS-box genes in flower formation. (please see the revised manuscript, Discussion section, Supplementary Fig. 3, 4)

REVIEWERS' COMMENTS:

Reviewer #1 (Remarks to the Author):

The authors have satisfactorily addressed my comments on the previous version. I also appreciate that in addition to the scientific content, they made extra effort to polish the languages, which read much more smoothly now.

The only minor comment I have on the revised version is the wording for the first sentence of the Abstract and the Introduction. For most species, the sepals are just uniformly green and do not contribute to interesting color patterns. Orchids are exceptional in that the sepals are as colorful as the petals. You could either make the statement more specific: "An unbalanced pigment distribution between the sepal and petal segments results in various colour patterns of ORCHID FLOWERS", or make it more general: "An unbalanced pigment distribution between different segments of the PERIANTH results in various FLOWER colour patterns".

Reviewer #2 (Remarks to the Author):

The authors almost resolved all the problems raised by the reviewers. I just have two minor points.

- 1) Line 279: should "Rhyncholaeliocattleya promoted carotenoid pigmentation 1" be RcPCP1?
- 2) Line 482: change "may regulate" into "may be regulated".

Reviewer #3 (Remarks to the Author):

I have no further comments. Good job!

Response to the reviewers' comments

Reviewer 1:

General comment: The authors have satisfactorily addressed my comments on the previous version. I also appreciate that in addition to the scientific content, they made extra effort to polish the languages, which read much more smoothly now.

The only minor comment I have on the revised version is the wording for the first sentence of the Abstract and the Introduction. For most species, the sepals are just uniformly green and do not contribute to interesting color patterns. Orchids are exceptional in that the sepals are as colorful as the petals. You could either make the statement more specific: "An unbalanced pigment distribution between the sepal and petal segments results in various colour patterns of ORCHID FLOWERS", or make it more general: "An unbalanced pigment distribution between different segments of the PERIANTH results in various FLOWER colour patterns".

Response: Thank you for your patience and suggestions. We have improved the Abstract and Introduction sections of the manuscript according to your advice.

Reviewer 2:

General comment: The authors almost resolved all the problems raised by the reviewers. I just have two minor points.

1) Line 279: should "Rhyncholaeliocattleya promoted carotenoid pigmentation 1" be RcPCP1?

2) Line 482: change "may regulate" into "may be regulated".

Response: Thank you for your patience and suggestions. We have improved these two place in our manuscript according to your advice.

Reviewer 3:

General comment: I have no further comments. Good job!

Response: Thank you for your patience.